# Vibrio deploys type 2 secreted lipase to esterify cholesterol with host fatty acids and mediate cell egress

Suneeta Chimalapati[1,2†], Marcela de Souza Santos[1†], Alexander E Lafrance[1], Ann Ray[3], Wan-Ru Lee[4], Giomar Rivera-Cancel[5], Gonçalo Vale[6], Krzysztof Pawlowski[1,7], Matthew A Mitsche[6,8], Jeffrey G McDonald[6,8], Jen Liou[4], Kim Orth[1,2,5*]

[1]Department of Molecular Biology, University of Texas Southwestern Medical Center, Dallas, United States; [2]Howard Hughes Medical Institute, University of Texas Southwestern Medical Center, Dallas, United States; [3]Department of Microbiology, University of Texas Southwestern Medical Center, Dallas, United States; [4]Department of Physiology, University of Texas Southwestern Medical Center, Dallas, United States; [5]Department of Internal Medicine, University of Texas Southwestern Medical Center, Dallas, United States; [6]Center for Human Nutrition, University of Texas Southwestern Medical Center, Dallas, United States; [7]Faculty of Agriculture and Biology, Warsaw University of Life Sciences, Warsaw, Poland; [8]Department of Molecular Genetics, University of Texas Southwestern Medical Center, Dallas, United States

*For correspondence:
kim.orth@utsouthwestern.edu

†These authors contributed equally to this work

**Abstract** Pathogens find diverse niches for survival including inside a host cell where replication occurs in a relatively protective environment. *Vibrio parahaemolyticus* is a facultative intracellular pathogen that uses its type 3 secretion system 2 (T3SS2) to invade and replicate inside host cells. Analysis of the T3SS2 pathogenicity island encoding the T3SS2 appeared to lack a mechanism for egress of this bacterium from the invaded host cell. Using a combination of molecular tools, we found that VPA0226, a constitutively secreted lipase, is required for escape of *V. parahaemolyticus* from the host cells. This lipase must be delivered into the host cytoplasm where it preferentially uses fatty acids associated with innate immune response to esterify cholesterol, weakening the plasma membrane and allowing egress of the bacteria. This study reveals the resourcefulness of microbes and the interplay between virulence systems and host cell resources to evolve an ingenious scheme for survival and escape.

## Introduction

As intracellular pathogens acquire mechanisms to invade a host cell, correlating mechanisms must also evolve for survival within the host and escape from the host. For example, the facultative intra-cellular pathogen *Vibrio parahaemolyticus* is a Gram-negative bacterium that resides in warm estua-rine environments with some strains acquiring virulence factors that can cause illness, even death in animals including shrimp and humans (*Wang et al., 2015*). This pathogen can cause acute gastroen-teritis due to the consumption of contaminated, undercooked seafood and possibly septicemia when infecting open wounds (*Wang et al., 2015*). *V. parahaemolyticus* contains a number of viru-lence factors, including hemolysins secreted via T2SS (Type 2 Secretion System) and two Type 3 Secretion Systems (T3SS1 and T3SS2) (*Makino et al., 2003*).

T2SS is primarily involved in exporting folded proteins from the periplasm of most Gram-negative bacteria into extracellular environment and is a part of the widely conserved general secretory (Sec) pathway (*Korotkov et al., 2012*; *Douzi et al., 2012*). T2SS is a specialized multicomponent assembly that consists of four major components: an outer membrane secretin, an inner membrane channel, the pseudopilus and an ATPase (*Douzi et al., 2012*; *Silva et al., 2020*). T2SS secreted protein repertoire includes various carbohydrate, lipid and protein hydrolyzing enzymes, pore-forming toxins, phosphatases, nucleases, etc. that are implicated in plant, animal and human pathogenesis and widely present in both intracellular and extracellular pathogens (*Nivaskumar and Francetic, 2014*; *Cianciotto and White, 2017*; *Cianciotto, 2005*). In *Vibrio* species, hemolysins including TDH (Thermostable Direct Hemolysin), TRH (TDH-related Hemolysin) and the cholera toxin are known to be secreted via the T2SS (*Matsuda et al., 2019*; *Sikora, 2013*).

Previous studies have shown that the more ancient T3SS1 is associated with all strains of *V. parahaemolyticus*, whereas T3SS2, a more recent acquisition, correlates with clinical isolates and disease in humans (*Makino et al., 2003*). To study the various virulence factors independently, deletions in particular genes are created to unmask the activity of a specific virulence mechanism (*De Souza Santos and Orth, 2019*). Herein, to study T3SS2, we utilize CAB2, a strain derived from the clinical isolate RIMD2210633 that is deleted for hemolysins and mutated for expression of the T3SS1 (*Zhang et al., 2012*).

The T3SS2, found on a pathogenicity island encodes a needle-like apparatus and effectors that mediate an invasive infection resulting in host cell death (*Zhang and Orth, 2013*; *de Souza Santos and Orth, 2014*). T3SS2 translocates the effector VopC, a deamidase, to mediate membrane ruffling and uptake of *V. parahaemolyticus* by nonphagocytic cells (*Zhang et al., 2012*; *de Souza Santos and Orth, 2014*). Once inside, *V. parahaemolyticus* escapes from an acidified endocytic compartment and proceeds to replicate in the cytoplasm of the host cell, reaching counts of 200–300 bacteria per host cell (*de Souza Santos and Orth, 2014*). Other translocated effectors have been shown to manipulate host cell signaling, including the acetyltransferase VopA that blocks MAPK signaling and the actin assembly factor VopL that blocks production of reactive oxygen species (*Trosky et al., 2004*; *Liverman et al., 2007*; *de Souza Santos et al., 2017*; *Trosky et al., 2007*). *V. parahaemolyticus* ultimately escapes from this protective replicative niche to infect other cells (*de Souza Santos and Orth, 2014*). In total, about a dozen T3SS2 effectors are thought to be delivered to the host cell, some with known molecular functions but with exception of the aforementioned effectors, understudied for their role in bacterial intracellular survival (*De Souza Santos and Orth, 2019*). After bioinformatic perusal of this pathogenicity island, there appeared to be no obvious candidate effector that would mediate the escape of *V. parahaemolyticus* from the endocytic compartment or the host cell.

To be a successful pathogen, an intracellular bacterium must egress after its replication in the host cell cytosol to re-infect neighboring cells and disseminate into tissues. Pathogens use various mechanisms for egress, including programmed cell death, non-lytic exit of host cells and manipulation of host-cell-derived membranes (*Hybiske and Stephens, 2015*; *Flieger et al., 2018*). Three forms of programmed cell death that include both non-lytic (apoptosis) and lytic pathways (pyroptosis and necroptosis) are observed in pathogen egress. For pathogen egress via apoptosis as seen with *M. tuberculosis* and *Leishmania* species, the invaded host cells are programmed to die without inducing inflammation. Thus, the pathogens cause less damage to the host leading to their dissemination within apoptotic bodies only to be engulfed by scavenging macrophages (*Martin et al., 2012*; *van Zandbergen et al., 2004*; *Peters et al., 2008*). Pyroptosis, induced by gasdermin in a caspase-dependent pathway, involves formation of pores in the plasma membrane and is used as an exit mechanism by *Francisella, Legionella, Shigella, Salmonella* and *Listeria* (*Hybiske and Stephens, 2008*; *Traven and Naderer, 2014*). Necroptosis, a programmed necrosis that is induced by the receptor-interacting protein kinase 3 (RIP3/RIPK3) and the mixed lineage kinase domain like pseudokinase (MLKL) signaling pathway, is observed for dissemination of *Salmonella* and *Mycobacterium* species (*Lindgren et al., 1996*; *Dallenga et al., 2017*). Following non-lysing pathways, many intracellular pathogens such as *Chlamydia, Rickettsia, Mycobacterium, Shigella, Listeria* and *Legionella* exit host cells by membrane protrusion, budding, exocytosis and expulsion (*Hybiske and Stephens, 2015*; *Flieger et al., 2018*; *Friedrich et al., 2012*) Parasites such as *Trypanosoma, Plasmodium* and *Toxoplasma gondi* use active host cell lysis mediated by proteases and lipases as exit strategy (*Hybiske and Stephens, 2015*; *Pszenny et al., 2016*).

Herein, we describe a novel mechanism deployed by *V. parahaemolyticus* for its egress from the host cells. We show that the lipase, VPA0226 is necessary for egress of *V. parahaemolyticus* from its host cell initially invaded in a T3SS2-mediated mechanism. However, VPA0226 is not secreted by the T3SS2, but rather it is constitutively secreted by the type two secretion system. Using lipidomic studies we found that, unlike other bacterial lipases, cytoplasmic VPA0226, but not its catalytically inactive forms (VPA0226-H/A or VPA0226-S/A), esterifies cholesterol using host polyunsaturated fatty acids (PUFA), specifically those implicated in immune signaling. Esterification of cholesterol leads to a weakened plasma membrane that allows *V. parahaemolyticus* to escape. Not only is this a novel mechanism that is using host-derived PUFA to esterify cholesterol for egress of an intracellular bacterial pathogen, this study exemplifies the resourcefulness and adaptability of bacteria to leverage an existing mechanism to survive and escape from a host cell.

## Results

### The catalytic activity of VPA0226 is required for *V. parahaemolyticus* egress from host cell

Intracellular pathogens are known to use lipases to escape from membrane compartments. SseJ, a T3SS lipase/esterase effector of *Salmonella* spp. is known to modulate the *Salmonella* containing vacuole lipid composition (*Ruiz-Albert et al., 2002*; *Fredlund and Enninga, 2014*). Using SseJ as a probe in a bioinformatic search of the *V. parahaemolyticus* genome, we identified a lipase (VPA0226) with approximately 16% sequence identity and conservation of catalytic residues (*Figure 1—figure supplement 1A*). Previously, in vitro studies demonstrated that VPA0226 has lipase activity and structural studies on a related *Vibrio vulnificus* lipase revealed a chloride-dependent catalytic diad (*Wan et al., 2019*; *Shinoda et al., 1991*). However, analysis of VPA0226 reveals a more classic catalytic triad of Ser-His-Asp (*Figure 1—figure supplement 1B*).

To test whether VPA0226 plays a critical role in *V. parahaemolyticus*' intracellular lifecycle, we performed a gentamicin protection assay, whereby HeLa cells were infected with the *V. parahaemolyticus* CAB2 strain for 2 hr to allow invasion of the host, followed by treatment with gentamicin for the remainder of the infection to kill all extracellular, but not intracellular bacteria (*de Souza Santos and Orth, 2014*). We observed that the *V. parahaemolyticus* CAB2 containing a functional T3SS2 can efficiently invade, replicate (indicated by the increase in colony-forming units (CFU)), and escape from the host cell (indicated by the after-peak decrease in CFU due to exposure of egressed bacteria to gentamicin) (*Figure 1A*, black bars). By contrast, a CAB2 strain deleted for *vpa0226* (CAB2Δ*vpa0226*) results in cells that have increasing amounts of bacteria with no significant decrease for up to 7 hr post-gentamicin treatment (PGT), indicating the inability of these bacteria to escape from an invaded cell (*Figure 1A*, green bars).

To test whether cell egress was attributed to VPA0226's catalytic activity, we complemented the CAB2Δ*vpa0226* with a wildtype copy of VPA0226 (CAB2Δ*vpa0226*+WT) or with a catalytically inactive copy of VPA0226 (CAB2Δ*vpa0226*+S/A). While in-trans complementation with a wildtype copy of VPA0226 allowed for bacterial escape, the catalytically inactive form of VPA0226 did not allow for egress of *V. parahaemolyticus* from the host cell. (*Figure 1B*). Furthermore, the decrease in number of lysed host cells, as well as the decreased number of egressed mutant bacteria at 7 hr PGT in comparison to its parental strain is consistent with these findings (*Figure 1—figure supplement 1C and D*, respectively). Similarly, when Caco-2 cells were infected with *V. parahaemolyticus* strains, CAB2Δ*vpa0226* or CAB2Δ*vpa0226*+S/A were unable to escape from invaded cells at 7 hr PGT, in contrast to the egress-competent CAB2 and CAB2Δ*vpa0226*+WT (*Figure 1—figure supplement 1E*). We next turned to confocal microscopy and observed over the course of an infection that only CAB2 and CAB2Δ*vpa0226*+WT, but not CAB2Δ*vpa0226* and CAB2Δ*vpa0226*+S/A, escape from the host cell 7 hr PGT (*Figure 1C and D*). These results support the previous observation that, in the absence of VPA0226, bacteria are unable to egress from their host cell. To further confirm that decrease in bacterial counts observed at 7 hr PGT in CAB2 or CAB2Δ*vpa0226*+WT invaded HeLa cells is not caused by gentamicin from the medium ultimately reaching intracellular bacteria, we performed an extended time course gentamicin protection assay in which HeLa cells were infected with *V. parahaemolyticus* strains for 2 hr, treated with gentamicin at 100 µg/ml for 1 hr and then changed to 10 µg/ml for the remainder of the assay. As observed previously, HeLa cells invaded with

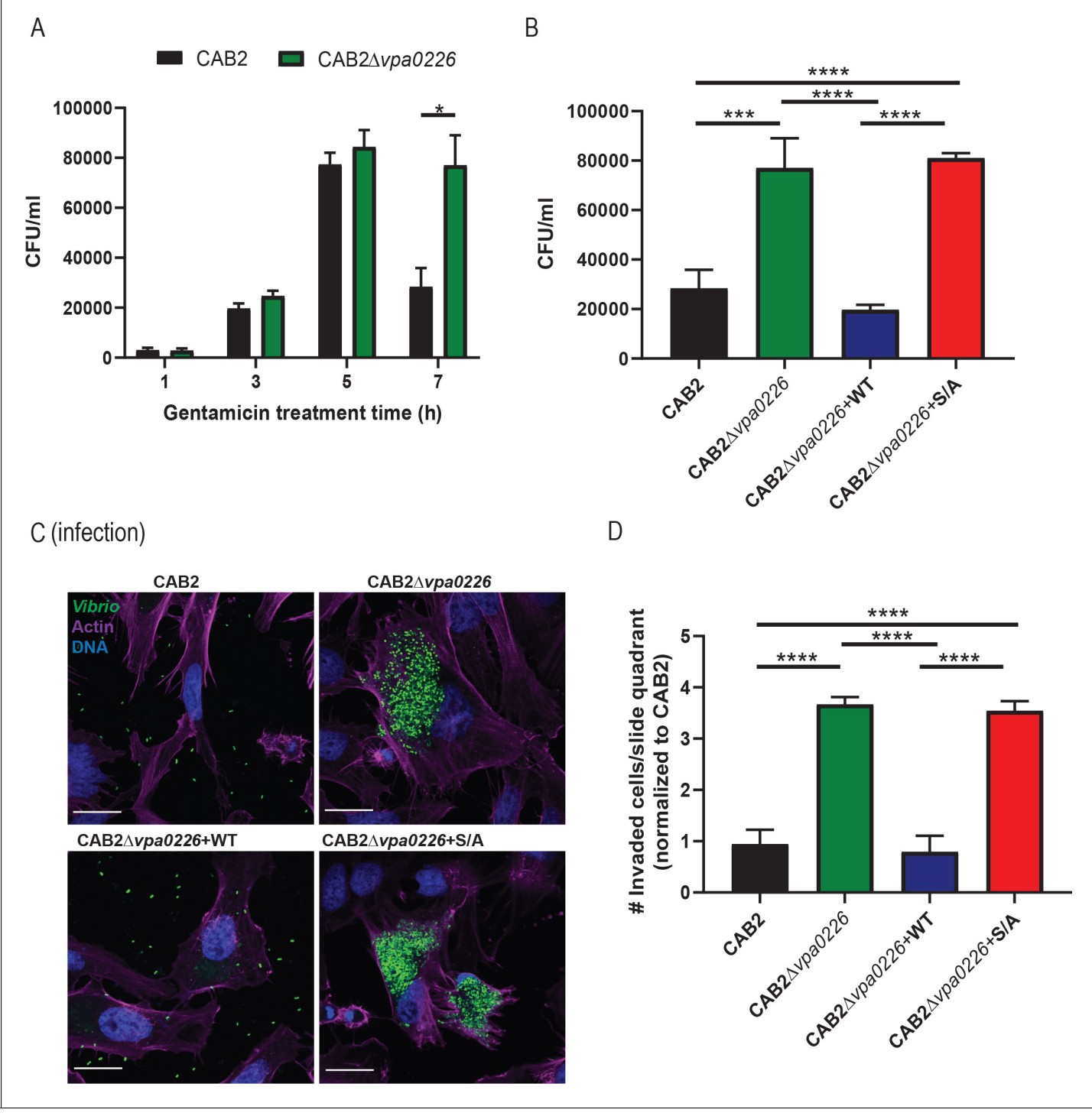

**Figure 1.** VPA0226 mediates bacterial egress from the host cell. (**A**) HeLa cells were infected with CAB2 or CAB2Δ*vpa0226* for 1, 3, 5, and 7 hr PGT. Host cell lysates were serially diluted and plated onto MMM agar plates for intracellular bacterial colony counting (CFU/mL). Numbers are expressed as an average of three technical replicates for one of three independent experiments. Error bars represent standard deviation from the mean. Asterisks represent statistical significance (*=p < 0.05) using two-way ANOVA and Turkey's multiple comparison test. (**B**) HeLa cells were infected with CAB2, CAB2Δ*vpa0226*, CAB2Δ*vpa0226*+WT, or CAB2Δ*vpa0226*+S/A for 7 hr PGT. Host cell lysates were serially diluted and plated onto MMM agar plates for intracellular bacterial colony counting (CFU/mL). Numbers are expressed as an average of three technical replicates for one out of three independent experiments. Error bars represent standard deviation from the mean. Asterisks represent statistical significance (***=p < 0.001, ****=p < 0.0001) using one-way ANOVA and Turkey's multiple comparison test. (**C**) Confocal micrographs of HeLa cells infected with GFP-expressing (green) CAB2, CAB2Δ*vpa0226*, CAB2Δ*vpa0226*+WT, or CAB2Δ*vpa0226*+S/A for 7 hr PGT. Host cell actin was stained with Alexa 680-phalloidin (magenta) and DNA

*Figure 1 continued on next page*

*Figure 1 continued*

was stained with Hoechst (blue). Scale bars = 25 μm. (D, relative to C) Quantification of HeLa cells containing intracellular bacteria per quadrant of slide. Numbers were normalized to CAB2 and are expressed as an average of three independent experiments. Error bars represent standard deviation from the mean. Asterisks represent statistical significance (****=p < 0.0001) using one-way ANOVA and Turkey's multiple comparison test.

The online version of this article includes the following figure supplement(s) for figure 1:

**Figure supplement 1.** VPA0226 is GCAT lipase that contributes to bacterial cell egress but does not contribute to *V. parahaemolyticus'* escape from its containing vacuole.

**Figure supplement 2.** Extended time course for gentamicin protection assay.

CAB2Δ*vpa0226* or CAB2Δ*vpa0226*+S/A strains still contained viable bacteria even at 12 hr PGT in contrast to CAB2 or CAB2Δ*vpa0226*+WT invaded HeLa cells that had no viable bacteria from 8 hr PGT onwards (*Figure 1—figure supplement 2A and B*). Interestingly, at 24 hr PGT, no viable bacteria were present in HeLa cells under any conditions and majority of the HeLa cells appeared rounded and dying (*Figure 1—figure supplement 2B*).

## VPA0226 is secreted by the type two secretion system

We initially speculated that VPA0226 is translocated as a T3SS effector, as it appears that VPA0226 is essential for *V. parahaemolyticus's* egress from the host cell when the bacterium leads a T3SS2-dependent intracellular life cycle (*Zhang et al., 2012*). To test this hypothesis, we followed the secretion of VPA0226 from various *V. parahaemolyticus* strains, including CAB2, CAB2Δ*vpa0226*, CAB3 (a strain that only contains the T3SS1), and CAB4 (a strain that contains neither T3SS1 nor T3SS2). To our surprise, with the exception of the CAB2Δ*vpa0226* strain, we observed VPA0226 to be constitutively secreted from all of these *V. parahaemolyticus* strains (*Figure 2A*). Further bioinformatic analysis using SignalP5.0 revealed that VPA0226 contained a signal peptide at its N-terminus similar to those found for enzymes that are secreted out of the type two secretion system (*Banerji et al., 2005*; *Figure 2—figure supplement 1*). To test whether VPA0226 is secreted through this system, we assayed the following strains for VPA0226 secretion: CAB2; CAB2Δ*vpa0226*; a CAB2 strain deleted for EpsD (an essential type two secretion system outer membrane component), CAB2Δ*epsD*; and the EpsD complemented strain CAB2Δ*epsD*+EpsD. We observed that VPA0226 is secreted by CAB2 and CAB2Δ*epsD*+EpsD, but not by CAB2Δ*vpa0226* or CAB2Δ*epsD* (*Figure 2B*). As we did not observe VPA0226 in the pellet fraction of CAB2Δ*epsD*, we performed qPCR analysis to confirm the expression of VPA0226 in CAB2Δ*epsD* strain and observed a significant reduction in the expression level of VPA0226 in CAB2Δ*epsD* compared to CAB2 (*Figure 2C*). In addition, when we overexpressed VPA0226 through a plasmid in CAB2Δ*epsD*, we observed a slightly higher migrating band corresponding to the uncleaved version of VPA0226 in the pellet fraction along with a few other lower migrating bands indicating possible degradation of VPA0226 (*Figure 2D*). Taken together, these above results demonstrate that VPA0226 is secreted by the type two secretion system (*Figure 2E*).

## Intracellular VPA0226 associates with and disrupts membranes of invaded cells

Since VPA0226 is predicted to be a lipase that hydrolyses fatty acids of phospholipids and can transfer fatty acids to cholesterol, we speculated that VPA0226 would be associated with a host membrane as it appears to be involved in bacterial egress from the host cell through this membrane. To test this hypothesis, we transiently transfected VPA0226 tagged with GFP (VPA0226-GFP) into HeLa cells and observed that cells expressing VPA0226-GFP died within 24 hr (data not shown); however, at an earlier time point (14 hr), the transfected cells were viable. Somewhat perplexing, VPA0226-GFP was not localized to the plasma membrane but to an intracellular membrane compartment (*Figure 3A*). To assess which compartment this might be, we co-stained transfected cells with markers for a number of intracellular compartments, including early endosomes using EEA1, late endosomes/lysosomes using Lamp-1, endoplasmic reticulum using calnexin and mitochondria using Acetyl-CoA acetyl transferase (ACAT1) and COXIV (*Figure 3*, *Figure 3—figure supplement 1A,B,C, D*). The only markers co-staining with VPA0226 were ACAT1 and COXIV, indicating that VPA0226 associates with the mitochondria (*Figure 3A*, *Figure 3—figure supplement 1D*). In addition, while

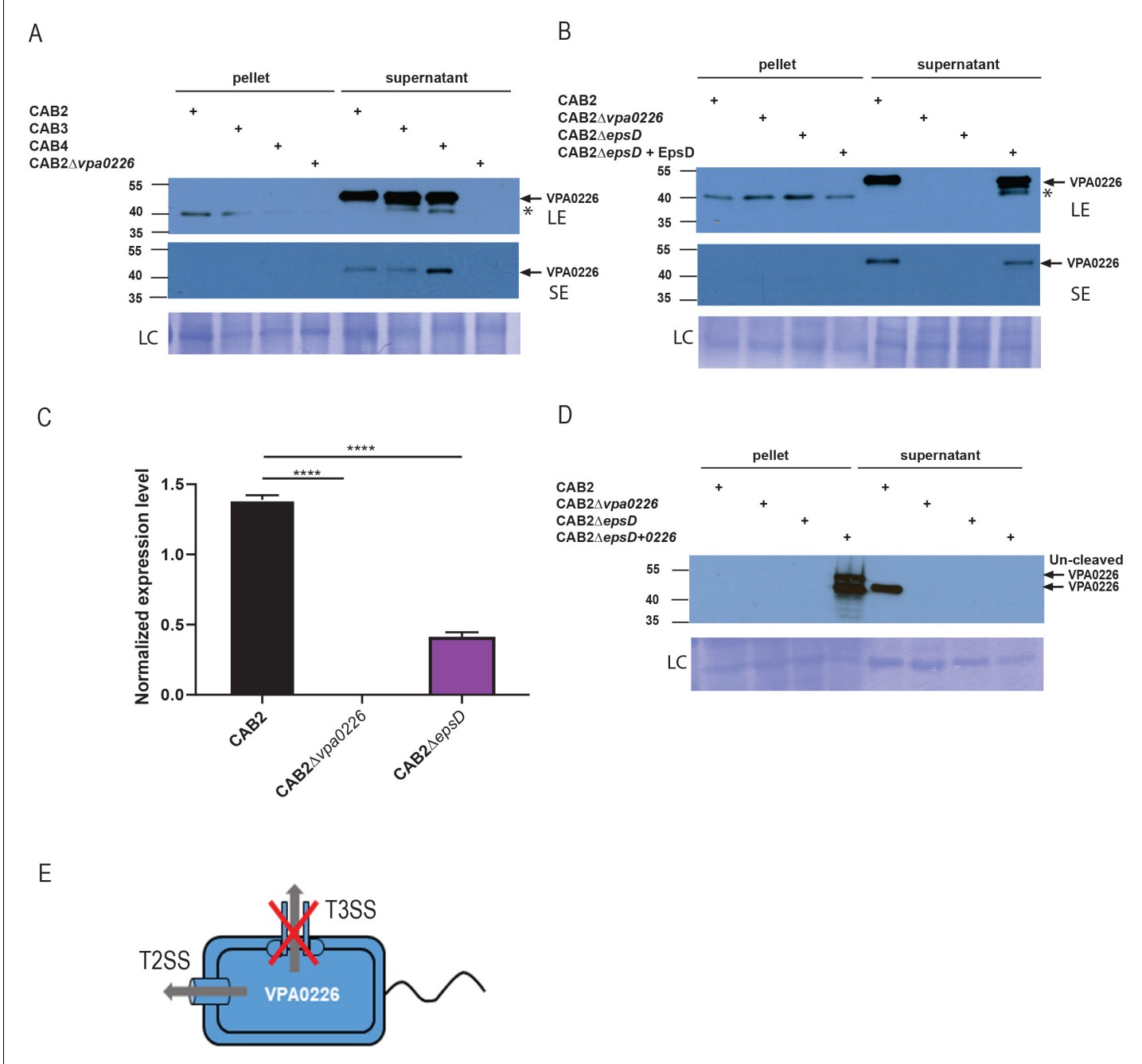

**Figure 2.** VPA0226 is a T2SS secreted protein. (A) Expression (pellet) and secretion (supernatant) of VPA0226 from CAB2, CAB3, CAB4 and CAB2Δ*vpa0226* detected by immunoblotting with anti-VPA0226 antibody. LC: loading control from total bacterial lysate or total secretion media, LE: Long exposure, SE: Short exposure and asterisk represents a variable non-specific band that disappears upon washing and short exposure. (B) Expression (pellet) and secretion (supernatant) of VPA0226 from CAB2, CAB2Δvpa0226, CAB2Δ*epsD* and CAB2Δ*epsD* + EpsD detected by immunoblotting with anti-VPA0226 antibody. LC: loading control from total bacterial lysate or total secretion media, LE: Long exposure, SE: Short exposure and asterisk represents a variable non-specific band that disappears upon washing and short exposure. (C) qPCR analysis for relative expression of VPA0226 in CAB2, CAB2Δvpa0226, CAB2Δ*epsD*. Expression was normalized to the house keeping gene RecA and asterisks represent statistical significance (****=p < 0.0001) using one-way ANOVA and Turkey's multiple comparison test. (D) Expression (pellet) and secretion (supernatant) of VPA0226 from CAB2, CAB2Δvpa0226, CAB2Δ*epsD* and CAB2Δ*epsD* + VPA0226 detected by immunoblotting with anti-VPA0226 antibody. LC: loading control from total bacterial lysate or total secretion media. (E) Schematic of VPA0226 secretion through T2SS and not T3SS2. The online version of this article includes the following figure supplement(s) for figure 2:

**Figure supplement 1.** Signal P-5 prediction of VPA0226.

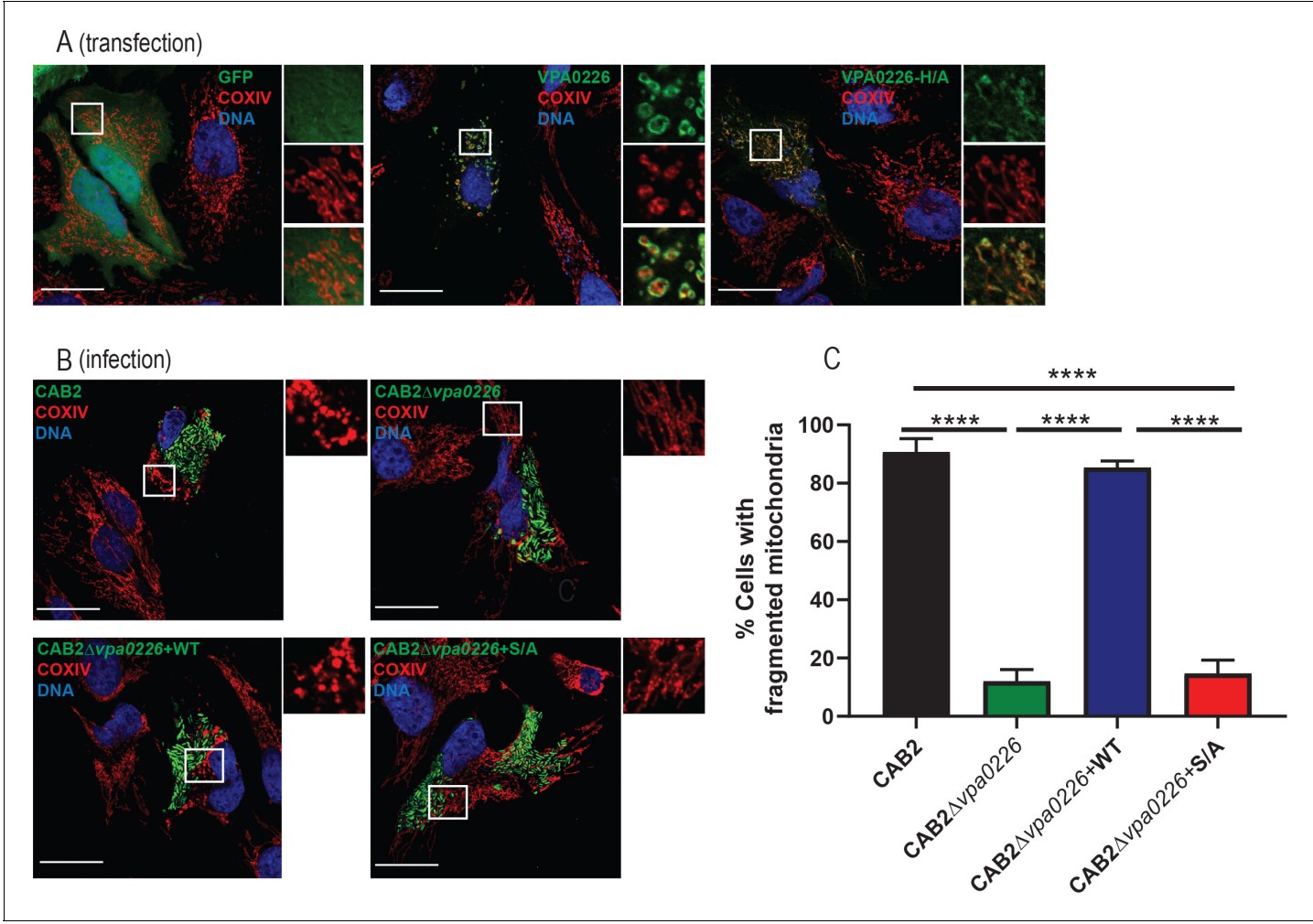

**Figure 3.** VPA0226 localizes to and fragments mitochondria in the host cell. (**A**) Confocal micrographs of HeLa cells transiently transfected with empty vector, VPA0226- or VPA0226-H/A-sfGFPN1 (green) for 14 hr. Mitochondria were stained with anti-COXIV antibody (red) and DNA was stained with Hoechst (blue). Scale bars = 25 μm. White boxes frame magnified areas. (**B**) Confocal micrographs of HeLa cells infected with GFP-expressing (green) CAB2, CAB2Δvpa0226, CAB2Δvpa0226+WT, or CAB2Δvpa0226+S/A. Mitochondria were stained with anti-COXIV antibody (red) and DNA was stained with Hoechst (blue) at 3.5 hr PGT. Scale bars = 25 μm. White boxes frame magnified areas. (C, relative to B) Quantification of HeLa cells exhibiting fragmented mitochondria. Numbers are expressed as an average of three independent experiments for 150 cells counted for each sample. Error bars represent standard deviation from the mean. Asterisks represent statistical significance (****=p < 0.0001) using one-way ANOVA and Turkey's multiple comparison test.

The online version of this article includes the following figure supplement(s) for figure 3:

**Figure supplement 1.** VPA0226 exclusively localizes to the mitochondria.

**Figure supplement 2.** Bacterially invaded cells display apoptotic signatures in a VPA0226-dependent manner.

mitochondria of cells mock-transfected or transfected with the catalytically inactive VPA0226 (VPA0226-H/A) show normal cell morphology, cells transfected with the wildtype VPA0226 display fragmented mitochondria (*Figure 3A*, *Figure 3—figure supplement 1D*).

The changes in morphology of the mitochondria also occurred during infection, as host cells invaded with the CAB2 strain revealed similar fragmented mitochondria at approximately 2 hr PGT (*Figure 3B and C*, *Figure 3—figure supplement 1E*). Cells that do not contain bacteria appear to have normal mitochondria (*Figure 3B and C*, *Figure 3—figure supplement 1E*). In addition, cells infected with CAB2Δvpa0226 strain exhibit normal mitochondrial morphology throughout the infection (*Figure 3B and C*, *Figure 3—figure supplement 1E*). Consistent with the transfection studies, mitochondrial fragmentation was dependent on the intact catalytic activity of VPA0226 (*Figure 3B and C*, *Figure 3—figure supplement 1E*). As we only saw changes in mitochondria when VPA0226

is provided by an intracellular source (invaded or transfected cells), we conclude that VPA0226 must be supplied intracellularly to cause changes in the mitochondria.

## *V. parahaemolyticus*-initiated apoptosis is not required for bacterial cell egress

Based on these studies, we made the supposition that disrupting lipids in the mitochondrial membrane causes fragmentation of this membrane and compromises the integrity of this compartment resulting in release of cytochrome C in the cytosol and the initiation of apoptosis. However, the initiation of these events would not support a mechanism for egress of *V. parahaemolyticus* from the host cell as apoptosis does not result in lysis or release of intracellular contents. Instead, we surmised if VPA0226 would compromise the cellular membranes by esterifying cholesterol, the integrity of all membranes would be compromised, including the plasma membrane, thereby allowing the escape of *V. parahaemolyticus* from host cells.

To address these hypotheses, we analyzed apoptotic signaling in host cells during infection. Since mitochondria are fragmented at 2 hr PGT, we tested whether Cytochrome C, an initiator of apoptosis, is released from the mitochondria at some point after this. Consistent with this idea, we observed the release of Cytochrome C into the cytosol in CAB2 and CAB2Δ*vpa0226*+WT-infected cells, but not CAB2Δ*vpa0226* and CAB2Δ*vpa0226*+S/A-infected cells (*Figure 3—figure supplement 2A,B*) by 5 hr PGT.

Based on the observations that Cytochrome C is released, we expected to observe downstream indicators of apoptosis, such as fragmented DNA. Indeed, we observed positive TUNEL staining in CAB2 and CAB2Δ*vpa0226*+WT infected cells but not in CAB2Δ*vpa0226* and CAB2Δ*vpa0226*+S/A infected cells after 5 hr PGT (*Figure 3—figure supplement 2C,D*). Interestingly, while treatment of cells with an inhibitor of apoptosis Z-VAD-FMK did inhibit apoptosis in staurosporine-treated cells, Z-VAD-FMK did not alter the progression of *V. parahaemolyticus*-mediated invasion or the survival and escape of the CAB2 strains from the host cells (*Figure 3—figure supplement 2E,F*). Based on this observation, we propose that VPA0226-mediated fragmented mitochondria lead to the initiation of apoptosis. However, death by apoptosis is not a mechanism used for cell lysis because *V. parahaemolyticus* escapes from the host cell 5 hr PGT whether or not apoptotic processes are active or inhibited (*Figure 3—figure supplement 2E*).

## VPA0226 esterifies cholesterol using host polyunsaturated fatty acids

To address how VPA0226-mediated fragmentation of the mitochondria was occurring, we analyzed the lipase activity of VPA0226 and its impact on host cellular lipids during the course of infection. Previously, it was shown that lipases similar to VPA0226 can esterify cholesterol by the transfer of an acyl group from an acyl-containing lipid to cholesterol thereby converting free cholesterol to a cholesteryl ester (*Buckley, 1982*; *Nawabi et al., 2008*). It should be noted that only the lipase activity, but not the transferase activity, has been observed previously for *Vibrio* lipases (*Shinoda et al., 1991*; *Van der Henst et al., 2018*). To confirm the lipase activity of VPA0226, we tested this enzyme with the EnzChek Phospholipase A2 (PLA2) Assay Kit (Invitrogen). Purified wildtype VPA0226 but not the catalytically inactive VPA0226- S/A mutant resulted in a concentration-dependent increase in fluorescence from a PLA2-specific fluorogenic substrate (BODIPY PC-A2) (*Figure 4—figure supplement 1A*). We next assessed whether VPA0226 can directly modify cholesterol by incubating purified secreted wildtype or catalytically dead VPA0226- S/A with liposomes containing lipid mixtures of cardiolipin and cholesterol, followed by lipid extraction and thin layer chromatography. We observed that the wildtype but not the catalytically inactive VPA0226-S/A mutant facilitated the esterification of cholesterol (*Figure 4—figure supplement 1B*).

We next asked whether the expression of VPA0226 would change the global profile of esterified cholesterol in cells. To test whether VPA0226 alters lipids in cells, we transfected HeLa cells with an empty vector, wildtype VPA0226 and the catalytically inactive VPA0226-H/A mutant. Our attempts to use the catalytically inactive VPA0226-S/A construct for mammalian cell expression was not successful. Therefore, we created a second construct where the catalytic histidine (residue 393) in the catalytic triad was mutated to alanine. This mutant VPA0226-H/A functions similar to the VPA0226-S/A mutant in the gentamicin protection assay (*Figure 4—figure supplement 2*). At 14 hr post-transfection, we performed lipidomic analysis of cells transfected with empty vector, VPA0226 and

the catalytically inactive VPA0226-H/A. Since cardiolipin is a lipid found exclusively in mitochondria and bacterial outer membranes and VPA0226, a constitutively secreted glycerophospholipid: cholesterol acyltransferase is localized to mitochondria, we speculated that VPA0226 might favor using fatty acids from cardiolipin (18:1,18:2,18:3) as a resource to esterify cholesterol. However, the levels of cholesteryl esters (CE) with cardiolipin fatty acids did not change significantly over the three samples (*Figure 4*). Rather, the cholesteryl esters that did dramatically change in the presence of wild-type VPA0226 included CE(20:3), CE(20:4), CE(20:5) and CE(22:6) (*Figure 4*). Interestingly, these fatty acids are ones implicated with innate immune signaling, with the most notable being arachidonic acid (20:4) (*Tallima and El Ridi, 2018*). As shown previously, depletion of cholesterol levels in cells lead to decrease in monolayer transepithelial electrical resistance and a significant increase in the paracellular permeability (*Lambert et al., 2005*; *Grunze and Deuticke, 1974*).

## Increase in esterified cholesterol compromises the integrity of the plasma membrane

We speculated that the dramatic increase in esterified cholesterol that correlated with the fragmented mitochondrial membranes would compromise the integrity of all cellular membranes, including the plasma membrane. To assess whether there was a decrease in free cholesterol, we tested whether the cellular sensor for free cholesterol, SREBP, was activated in cells transfected with

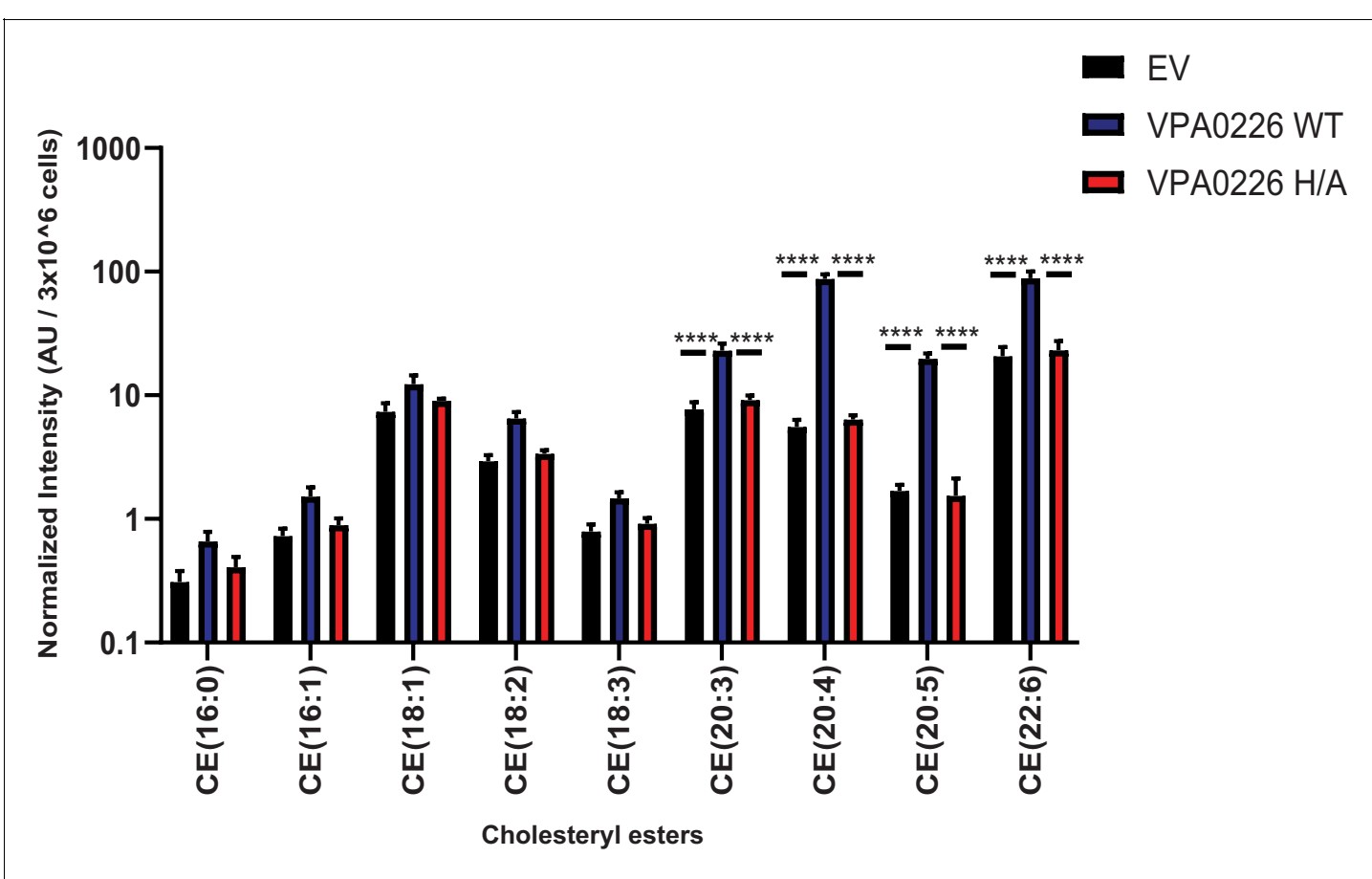

**Figure 4.** VPA0226 changes lipid profile in cells. Quantification of polyunsaturated and saturated fatty acyl cholesteryl esters extracted from transfected HeLa cells (empty vector (EV), wildtype VPA0226-sfGFPN1 and VPA0226-H/A-sfGFPN1). Numbers are expressed as an average of five technical replicates for one of three independent experiments. Error bars represent standard deviation from the mean. Asterisks represent statistical significance (****=p < 0.0001) using one-way ANOVA and Turkey's multiple comparison test.

The online version of this article includes the following figure supplement(s) for figure 4:

**Figure supplement 1.** VPA0226 displays phospholipase 2 (PLA 2) activity.

**Figure supplement 2.** CAB2Δ*vpa0226*+H/A mutant displays an egression defect similar to CAB2Δ*vpa0226* and CAB2Δ*vpa0226*+S/A.

VPA0226 (*Brown and Goldstein, 1997*). HeLa cells were transfected for 14 hr (the same time point used with lipidomic assays described above) with empty vector, VPA0226 or the catalytically inactive VPA0226H/A and qPCR analysis was performed assessing genes that are targets of SREBP transcriptional activity. Using qPCR, we observed all three genes, HMGCR, HMGCS and DHCR24, were upregulated in cells expressing VPA0226, but not in cells with the empty vector or the catalytically inactive VPA0226H/A (*Figure 5A*). Further, the plasma membrane fractions isolated from HeLa cells expressing VPA0226 had a statistically significant reduction in free cholesterol content compared to cells expressing either empty vector or catalytically inactive VPA0226 (*Figure 5B*, *Figure 5—figure supplement 1A,B*). These data support the proposal that free cholesterol levels are being compromised in cells expressing VPA0226.

Next, we asked whether cells subjected to VPA0226 activity might be more fragile when subjected to mechanical stress, in this case we used osmotic stress. HeLa cells were transfected for 14 hr with empty vector, VPA0226 or the catalytically inactive VPA0226-H/A and then treated with water containing propidium iodide to assess their susceptibility to hypotonic stress. We observed that cells expressing VPA0226, but not in cells with the empty vector or the catalytically inactive VPA0226-H/A, lysed more readily as indicated by an increase in fluorescence (*Figure 5C*).

To further assess the integrity of the plasma membrane during infection, HeLa cells were infected with CAB2 strains and tested for permeability using Sytox, a stain that is only permeable to damaged membranes. We observed that cells infected with either CAB2 or CAB2Δ*vpa0226*+WT, but not with CAB2Δ*vpa0226* and CAB2Δ*vpa0226*+S/A, take up the Sytox stain at 5 hr PGT, indicating that their membranes have been compromised (*Figure 5D and E*). Furthermore, only cells that are invaded with CAB2 take up the Sytox stain, consistent with the observation that only the invaded cells with an internal source of VPA0226 have compromised membranes (*Figure 5—figure supplement 1C*).

Finally, to confirm VPA0226 facilitates egress of *V. parahaemolyticus* by changing levels of cholesterol in infected cells, we treated CAB2 and CAB2Δ*vpa0226*-infected host cells with 100 µM cholesterol resulting in the addition of free cholesterol to plasma membrane (*Chakrabarti et al., 2017*). We discovered that CAB2 was no longer able to efficiently egress from cholesterol-treated cells at 7 hr PGT and there was no change in the inability of CAB2Δ*vpa0226* to egress from host cells (*Figure 5F and G*). We then treated CAB2 and CAB2Δ*vpa0226* infected cells with hydroxypropyl β-cyclodextrin (HPCD), a compound that depletes cholesterol from membranes (*Chakrabarti et al., 2017*). We observed that CAB2Δ*vpa0226* could now escape from host cells at 7 hr PGT and that CAB2 followed an expected course for egress, albeit, somewhat accelerated (*Figure 5F and H*). These studies support our model that changes in free cholesterol are essential for egress of *V. parahaemolyticus* from host cells.

## Discussion

Intracellular bacteria must find mechanisms to invade, survive, and escape the host cell. Previously, the mechanism of entry for *V. parahaemolyticus* was discovered to be mediated by the deamidase activity of VopC (*Zhang et al., 2012*). Further studies uncovered mechanisms used by *V. parahaemolyticus* to attenuate signaling pathways inside the infected cell, by effectors such as VopA and VopL (*De Souza Santos and Orth, 2019*; *de Souza Santos et al., 2017*). Many other mechanisms remain to be elucidated for *V. parahaemolyticus*'s defense against the host. Herein, we find that the escape route for *V. parahaemolyticus* mediated host cell invasion is dependent on the Type two secreted lipase VPA0226, which compromises the integrity of the plasma membrane by using polyunsaturated fatty acids to esterify cholesterol in the infected cell. For this pathogen, VPA0226 provides an essential activity for the virulence of the T3SS2 pathogenicity island by providing a convenient mechanism for escape. When dissecting the activity mediated by VPA0226, we were struck by its many unconventional and somewhat perplexing properties, including its secretion, enzymology and biology.

VPA0226 is secreted, not by the T3SS2, but by the type two secretion system. These findings make it very clear that, although we attribute *V. parahaemolyticus* virulence to T3SS2 (11,43), this bacterial pathogen uses many tools outside of this pathogenicity island for a successful infection. Other tools would include external host signaling factors, such as bile salts for activation of the system and adhesion proteins used to attach to host cells (*Li et al., 2016*; *Krachler et al., 2011*). Another example supporting notion is the fact that the *V. parahaemolyticus* T3SS1 uses an effector,

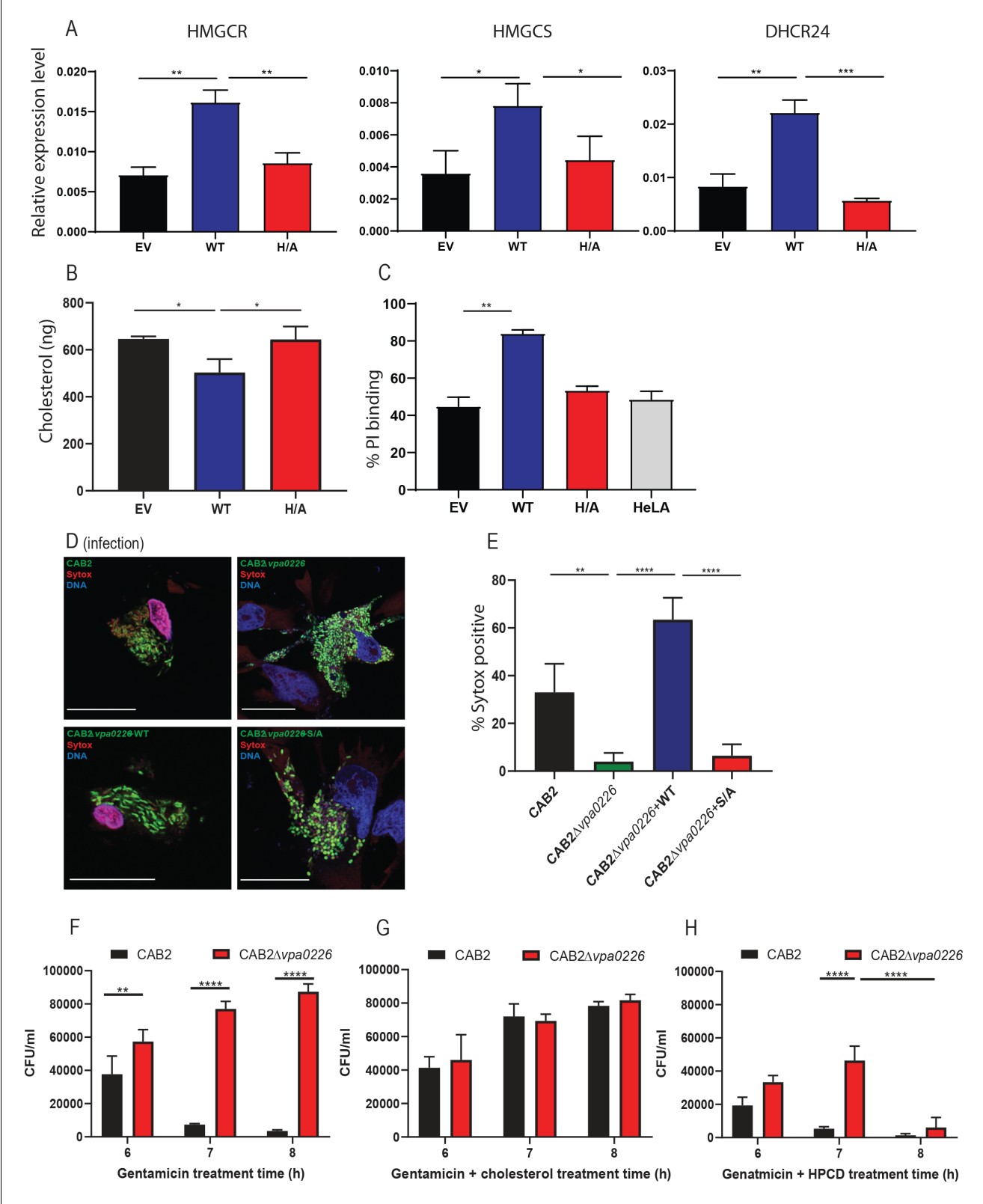

**Figure 5.** VPA0226 modifies cholesterol content in the plasma membrane, upregulates cholesterol synthesis gene expression in cells and permeabilizes the host cell plasma membrane. (**A**) qPCR analysis for relative expression of HMGCR, HMGCS and DHCR24, signature genes involved in cholesterol synthesis (SREBP2) pathway in HeLa cells transfected with empty vector (EV), wildtype VPA0226-sfGFPN1 and VPA0226-H/A-sfGFPN1. Expression was normalized to the house keeping gene GAPDH and asterisks represent statistical significance (*=p < 0.05, **=p < 0.01, ***=p < 0.001) using one-way

*Figure 5 continued*

ANOVA and Turkey's multiple comparison test. (B) Quantification of cholesterol in plasma membranes isolated from HeLA cells transfected with empty vector (EV), wildtype VPA0226-sfGFPN1 and VPA0226-H/A-sfGFPN1. Asterisks represent statistical significance (*=p < 0.05) using one-way ANOVA and Turkey's multiple comparison test. (C) Susceptibility to mechanical stress as measured by propidium iodide (PI) binding in transfected HeLA cells (empty vector (EV), wildtype VPA0226-sfGFPN1 and VPA0226-H/A-sfGFPN1), **=p < 0.01 using one-way ANOVA and Turkey's multiple comparison test. (D) Confocal micrographs of HeLa cells infected with GFP-expressing (green) CAB2, CAB2Δvpa0226, CAB2Δvpa0226+WT, or CAB2Δvpa0226+S/A at 5 hr PGT. Cell permeability was assessed by staining with dye sytox (red), bacteria (green), and DNA (blue, stained with Hoechst). Scale bars = 25 μm. Arrows point to the sytox permeable HeLA cell nuclei in CAB2, and CAB2Δvpa0226+WT, or sytox impermeable nuclei in CAB2Δvpa0226, CAB2Δvpa0226+S/A infections. (E, relative to D) Quantification of HeLa cells positive for Sytox staining. Numbers are expressed as an average of three independent experiments for 150 cells counted for each sample. Error bars represent standard deviation from the mean. Asterisks represent statistical significance (**=p < 0.01, ****=p < 0.0001) using one-way ANOVA and Turkey's multiple comparison test. (F, G and H) HeLa cells were infected with CAB2 or CAB2Δvpa0226 for 6, 7 and 8 hr PGT along with either cholesterol/MCD or HPCD or without. Host cell lysates were serially diluted and plated onto MMM agar plates for intracellular bacterial colony counting (CFU/mL). Numbers are expressed as an average of three technical replicates for one of three independent experiments. Error bars represent standard deviation from the mean. Asterisks represent statistical significance (**=p < 0.01, ****=p < 0.0001) using 2way ANOVA and Turkey's multiple comparison test.

The online version of this article includes the following figure supplement(s) for figure 5:

**Figure supplement 1.** Subcellular fractionation and sytox staining of HeLa cells infected with GFP-expressing (green) CAB2.

VPA0450, located outside its pathogenicity island to mediate an orchestrated host cell death (*Broberg et al., 2011*; *Broberg et al., 2010*).

VPA0226 appears to be constitutively secreted from *V. parahaemolyticus* by the type two secretion system, as we saw no accumulation of VPA0226 inside *V. parahaemolyticus* cells, indicating that, as it is synthesized it is secreted (*Figure 2*). Additionally, when media is collected from log-phase *V. parahaemolyticus* cultures, we were able to purify VPA0226 from the culture medium. We propose that VPA0226 is constitutively secreted by intracellular bacteria during infection because we only observe changes in the mitochondria, release of cytochrome C, TUNEL-positive cells and Sytox membrane permeability in host cells that are invaded with *V. parahaemolyticus* expressing wild-type VPA0226. Therefore, VPA0226 targets host cell machinery from within the host cell and not from the outside.

With regard to its localization, we observe transfected VPA0226 localized to the mitochondria and during *invasion* of *V. parahaemolyticus* and *transfection* o f VPA0226, the mitochondria are fragmented. Neighboring cells that are not invaded or transfected appear to have mitochondria with a normal phenotype. Although we have been unable to detect VPA0226 during an infection, we propose using molecular Koch's postulate that the wildtype but not the catalytically inactive VPA0226 causes the fragmented mitochondrial phenotype (*Falkow, 1988*). As VPA0226 does not appear to encode a mitochondrial localization domain, how and why it localizes specifically to the mitochondria is unclear.

Secreted phospholipases have been associated with many virulence activities, including mucus degradation, hemolysis and phagosomal escape (*Flores-Díaz et al., 2016*). These enzymes have broad substrate specificity and can work both as lipases and acyl transferases. Homologues of VPA0226 in *Vibrio* species such as *vhhA* in *V. harveyi*, *VvPlpA* in *V. vulnificus*, and *phlA* in *V. mimicus*, are implicated in virulence but have only been shown to possess the lipase activity (*Zhong et al., 2006*; *Jang et al., 2017*; *Lee et al., 2002*). Recombinant VvhA from *Vibrio vulnificus* was shown to induce NF-κB-dependent mitochondrial cell death mediated by the production of lipid raft-dependent ROS (*Lee et al., 2015*). Another structural homologue *lec/VC_A0218* in *V. cholerae* with lipase activity is implicated in escape from the infected amoeba (*Van der Henst et al., 2018*). In addition, a genetically non-homologous but functionally similar TgLCAT, lecithin:cholesterol acyltransferase encoded by a protozoan parasite *Toxoplasma gondii* is also implicated in host cell egress (*Pszenny et al., 2016*).

Using bioinformatics and clustering analysis, we find that VPA0226 is in a clan of enzymes containing invariant catalytic Ser, Gly, Asn and His residues which are referred to as GDSL-like lipases (*Akoh et al., 2004*; *Figure 1—figure supplement 1B*). VPA0226 and homologs from other *Vibrio* species, together with the *Salmonella* phospholipase effector SseJ, are located in a mostly bacterial subfamily, closely related to plant and fungal GDSL lipases and only remotely related to animal GDSL lipases, for example human Phospholipase B1 (*Figure 4—figure supplement 1C*). Based on

this conservation, we predict that the presence of VPA0226 in the *V. parahaemolyticus* genome predated the acquisition of the T3SS2 pathogenicity island (*Makino et al., 2003*). From an evolutionary standpoint, it is interesting (and resourceful on the part of a bacterial pathogen) that this enzyme subsequently became an essential factor for escape of the invading *V. parahaemolyticus* from cells. Interestingly, the bacterial-plant-fungal GDSL lipase subfamily includes a large plant group of extracellular lipases (e.g. more than 100 different proteins in *Arabidopsis*), many of which are involved in regulation of defense against bacterial and fungal pathogens. Fungal members of the subfamily are found in most fungal lineages, including several pathogenic, but excluding yeast-like ones. Majority of this subfamily members are predicted to be secreted by possessing a classical signal peptide while a number of bacterial members additionally possess the autotransporter 'translocator' domains (*Benz and Schmidt, 2011*). This may suggest an alternative, speculative evolutionary scenario for this subfamily, involving horizontal gene transfers between pathogens and hosts.

Our work confirms previous observations demonstrating that VPA0226 has lipase activity (*Figure 4—figure supplement 1A*; *Shinoda et al., 1991*). We went on to show both in vitro and upon ectopic expression of VPA0226 in mammalian cells VPA0226 can function as an acyl transferase. Interestingly, VPA0226 appears to act like a phospholipase A2 (PLA2) because, using lipidomics, we observe that VPA0226 specifically transfers an acyl group from position two from the glycerol backbone of a phospholipid to cholesterol (*Haas and Stanley, 2007*). The products from this reaction are esterified cholesterol and lysophospholipids, both of which destabilize membranes (*Aikawa et al., 2015*). The esterified cholesterol produced by VPA0226 are quite selective PUFAs, including 20:3, 20:4 (arachidonic acid) and 20:5 and 22:6 (fish oil lipids). This result was quite surprising for two reasons: first, the GDSL-like lipase from bacteria have been shown to use fatty acids (such as cardiolipin) as substrates as these are associated with bacteria; second, although eukaryotes produce cardiolipin, the substrate of choice for VPA0226 was clearly to use the aforementioned PUFAs to esterify cholesterol. Interestingly, these PUFAs (arachidonic acid and fish oil lipids) are implicated with immune signaling (*Tallima and El Ridi, 2018*; *Nicolaou et al., 2014*). Recently, it has been reported that *L. monocytogenes* and *Shigella flexneri*, manipulate the accessible pool of cholesterol for infection and that acute mobilization of this plasma membrane cholesterol pool by oxysterols confers immunity (*Abrams et al., 2020*).

From a biological perspective, we observe a well-timed, orchestrated escape of *V. parahaemolyticus* from the host cell. After *V. parahaemolyticus* escapes from the endosome, we postulate VPA0226 is secreted into the host cytosol, localizes to the mitochondria and esterifies cholesterol over the course of about 5–6 hr. During that time, *V. parahaemolyticus* reaches a density of about 300 bacteria/invaded cell and, due to mechanical stress (or possibly another unknown factor) egresses from the host cell. We observe that *V. parahaemolyticus* uniquely utilizes two independent secretion systems to support its intracellular lifestyle. From the perspective of virulence, *V. parahaemolyticus* appears to have utilized existing resources to evolve an efficient mechanism for invasion, propagation and escape. We speculate, considering the time pathogens have to evolve, that there is actually more interplay between secretion systems, providing unique, synergistic mechanisms to support a successful lifestyle of possibly pathogenicity, symbiosis and/or parasitosis.

## Materials and methods

### Bacterial strains, cell lines and culture conditions

The *V. parahaemolyticus* CAB2 strain was derived from POR1 (clinical isolate RIMD2210633 lacking TDH toxins), the latter being a generous gift from Drs. Tetsuya Iida and Takeshi Honda (*Zhang et al., 2012*; *Park et al., 2004*). The CAB2 strain was made by deleting the gene encoding the transcriptional factor ExsA, which regulates the activation of the T3SS1 (*Zhang et al., 2012*; *Zhou et al., 2010*). CAB2 was grown in Luria-Bertani (LB) medium, supplemented with NaCl to a final concentration of 3% (w/v), at 30℃. When necessary, the medium was supplemented with 50 μg/mL spectinomycin (to select for growth of CAB2-GFP strains [*Ritchie et al., 2012*]) or 250 μg/mL kanamycin. HeLA and Caco-2 cell lines have been obtained from ATCC, which have been authenticated and tested to confirm the absence of mycoplasma contamination.

## Antibodies

Antibodies were purchased from following companies. Rabbit polyclonal anti LAMP-1 –Abcam, mouse polyclonal anti-EEA1 and mouse monoclonal anti Cytochrome C – BD Biosciences, rabbit monoclonal anti COXIV and rabbit monoclonal anti-calnexin – Cell Signaling Technologies, rabbit polyclonal anti-ACAT1 – Genetex, rabbit polyclonal anti-Histone H3 – Millipore Sigma, goat polyclonal anti-E-Cadherin – R and D Systems, and mouse monoclonal anti GAPDH – Invitrogen. Secondary antibodies Alexa Fluor 647/568 goat anti-mouse/rabbit were obtained from Life Technologies, and Alexa Fluor 555 goat anti-mouse from Invitrogen. The antibody against VPA0226 was custom-made by Thermo Scientific.

## Deletion of *vpa0226* and *epsD* from CAB2 strain

For in-frame deletion of *vpa0226* (GeneBank sequence accession number NC_004605.1), the nucleotide sequences 1kb upstream and downstream of the gene were cloned into pDM4, a Cm$^r$ Ori6RK suicide plasmid (12). Primers used were 5' GATCGTCGACATCAAATTGAATGCACTATGATC 3' and 5' GATCACTAGTAAAGAAGACCCCTTTATTGATTC 3' for amplification of 1kb upstream region, and 5' GATCACTAGTCTAGCGAGCACATAAAAAGC 3' and 5' GATCAGATCTTCCGGGGTGG TAAATGCTT 3' for 1kb downstream region. 1kb sequences were then inserted between SalI and SpeI sites (upstream region) or SpeI and BglII (downstream region) sites of the plasmid multiple cloning site. For the deletion of *epsD*, the primers used were 5' GATCGTCGACATCAAATTGAATGCAC TATGATC 3' and 5' GATCACTAGTAAAGAAGACCCCTTTATTGATTC 3' for amplification of 1kb upstream region, and 5' GATCACTAGTCTAGCGAGCACATAAAAAGC 3' and 5' GATCAGATC TTCCGGGGTGGTAAATGCTT 3' for 1kb downstream region. 1kb sequences were then inserted between SalI and SpeI sites (upstream region) or SpeI and BglII (downstream region) sites of the plasmid multiple cloning site. The resulting construct was inserted into CAB2 via conjugation by S17-1 (λ*pir*) *Escherichia coli*. Transconjugants were selected for on minimal marine medium (MMM) agar containing 25 µg/mL chloramphenicol. Subsequent bacterial growth on MMM agar containing 15% sucrose (w/v) allowed for counter selection and curing of *sacB*-containing pDM4. Deletion was confirmed by PCR and sequencing analysis.

## Complementation of CAB2Δ*vpa0226* and CAB2Δ*epsD*

For reconstitution of CAB2Δ*vpa0226*, the sequence coding for *vpa0226* was amplified using primers 5' GATCCTGCAGATGCTTAAAATTAAACTGCCT 3' and 5' GATA GAATTCTTACTTATCGTCGTCA TCCTTGTAATC 3' and then cloned into the pBAD/*Myc*-His vector (Invitrogen, resistance changed from ampicillin to kanamycin) between PstI and EcoRI sites. For reconstitution of CAB2Δ*epsD* full length EpsD along with 1.2kb upstream region was amplified using primers 5' ATCGGAGCTCA-GAACACTACGCTATCTAATCACTG 3' and 5' ATCGGGTACCGTTCTTTTCCATTTGATCGAT 3' and cloned into SacI and KpNI sites of pBAD/*Myc*-His vector. The resulting construct was inserted into CAB2Δ*vpa0226* or CAB2Δ*epsD* via triparental conjugation using *E. coli* DH5α (pRK2073). Transconjugants were selected for on MMM agar containing 250 µg/mL kanamycin. Reconstitution was confirmed by PCR. Empty pBAD plasmid (without *vpa0226/epsD* gene insertion) was introduced to CAB2 and or CAB2Δ*epsD* strains for consistency in bacterial strain manipulation. Expression of *vpa0226* was induced by bacterial culturing in the presence of 0.02% arabinose to activate pBAD's araC promoter and *EpsD* was expressed through its native promoter. For the expression of catalytically inactive VPA0226, a serine residue at position number 153 was modified to alanine through site directed mutagenesis using VPA0226pBAD/*Myc*-His vector as a template and primers 3' CCAACG TGAGCCACTGCGGAACAGACTATGTCCG 5' and 5' GGTTGCACTCGGTGACGCCTTGTCTGA TACAGGC 3'. The resulting construct was inserted into CAB2Δ*vpa0226* as detailed above.

## T3SS2 and T2SS expression and secretion assay

For the expression and secretion of T3SS2, overnight grown *V. parahaemolyticus* strains in MLB at 30°C were diluted to OD600 of 0.3 in LB supplemented with 0.05% bile salts and grown at 37°C for 3 hr, as previously described (*Gotoh et al., 2010*). For the expression fraction, bacterial cultures of OD600 of 0.5 were pelleted and resuspended in 2x Laemmli buffer. For the secretion fraction, bacterial culture supernatants were filtered with a 0.22micron filter and precipitated with deoxycholate (150 µg/ml) and trichloroacetic acid (7% v/v). Precipitated proteins were pelleted, washed twice with

acetone and then resuspended in 2x Laemmli buffer. Expression and secretion levels were detected by western blot analysis. Total protein load was assessed by staining the nitrocellulose membrane with Coomassie blue. For T2SS expression and secretion, the same protocol was used except that the cultures were grown at 30°C in MLB without the addition of bile salts.

## Lactate dehydrogenase (LDH) cytotoxicity assay

HeLa cells were plated in triplicate in a 24-well tissue culture plate at $7 \times 10^4$ cells per well and grown for 16–18 hr. Overnight grown bacterial cultures in MLB at 30°C were diluted to OD600 of 0.3 in MLB supplemented with 0.05% bile salts and grown at 37°C for 90 min to induce T3SS2. Induced *V. parahaemolyticus* cultures were then used to infect HeLa cells at MOI 10. Immediately after the addition of bacteria to HeLa cells, plates were centrifuged for 5 min at 1000 rpm to synchronize infection. Gentamicin was added to HeLa cells at 100 μg /ml after 2 hr of infection. Seven hours post PGT, 200 μL of media from each well was transferred to a 96-well plate and centrifuged at 1000 rpm for 5 min, then 100 μL of this supernatant was used to evaluate LDH release into the culture medium as a measure of host cells lysis using a colorimetric cytotoxicity detection kit (Takara Bio) according to manufacturer's instructions. HeLa cells treated with 1% Triton X-100 for 10 min at the end of gentamicin treatment were used as positive control and cell lysis was expressed as a % of Triton-treated cell lysis.

## Gentamicin protection assay

Bacteria were added to triplicate wells of HeLa cell monolayers for infection as described in LDH assay. Caco2 cells were plated at $2 \times 10^5$ cells/ml in 24 wells. All infections were carried out at a MOI of 10. Gentamicin was added at 100 μg/mL to each well after 2 hr of infection to kill extracellular bacteria. At each indicated time point, monolayers of HeLa cells were washed with PBS and cells were lysed by incubation with PBS containing 0.5% Triton X-100 for 10 min at room temperature with agitation. Serial dilutions of lysates were plated on MMM plates and incubated at 30°C overnight for subsequent CFU enumeration. Z-VAD-FMK (EMD Millipore) was added to HeLa cells at the beginning of infection at 50 μM where indicated. Z-VAD-FMK at this concentration prevented apoptotic cell-death induced by 1.2 μM staurosporine (Abcam). When needed HeLa cells were treated with either 100 μM Cholesterol (Sigma) along with a carrier, randomly methylated β-cyclodextrin (MCD) (Cyclodextrin Technologies Inc) to increase membrane cholesterol levels or 1% (w/v) hydroxypropyl β-cyclodextrin (HPCD) (Cyclodextrin Technologies Inc) to deplete cholesterol along with gentamicin treatment.

## Immunofluorescence and confocal microscopy

HeLa cells were seeded onto six-well plates containing sterile coverslips at a density of $7 \times 10^4$ cell/mL. Following infections with *V. parahaemolyticus* strains and indicated gentamicin treatment times, cells were washed with PBS and fixed in 3.2% (v/v) para-formaldehyde for 10 min at room temperature. Fixed cells were washed in PBS and permeabilized with either 0.5% Triton X-100 in PBS (EEA1) or 0.5% saponin in PBS (LAMP1) for 4 min at room temperature (*de Souza Santos and Orth, 2014*). For calnexin, ACAT1, COXIV and Cyto C, fixed cells were permeabilized with ice cold methanol for 10 min at −20°C. After washing with PBS, blocking was performed with either 1% BSA (EEA1), 1% BSA with 0.1% saponin (LAMP1) or 5% normal goat serum with 0.3% Triton X-100 (calnexin, ACAT1, COXIV and Cyto C) for 30 min at room temperature. Primary and secondary antibodies were diluted in PBS containing either 0.5% BSA and 0.25% Tween 20 (EEA1), 0.5% BSA and 0.25% saponin (LAMP1) or 1% BSA and 0.3% Triton X-100 at appropriate dilutions and incubated serially for 1 hr at room temperature. Between each antibody incubation, coverslips were washed three times with PBS or PBS containing 0.1% saponin (LAMP1) for 5 min each wash. Nuclei and actin cytoskeleton were stained with Hoechst (Sigma) and rhodamine-phalloidin (Molecular Probes), respectively. For Sytox orange (Thermo Fisher Scientific) staining, cells were incubated with 0.5 μM Sytox orange for 15 min following infections with *V. parahaemolyticus* strains and gentamicin treatment. TUNEL staining was performed using DeadEnd fluorometric TUNEL system (Promega) according to manufacturer's instructions. Briefly, HeLa cells following infections and gentamicin treatment were fixed with 3.2% paraformaldehyde for 10 min at room temperature and permeabilized using 0.2% triton x 100 in PBS for 5 min at room temperature. Cells were then incubated with equilibration buffer for 10 min at

room temperature followed by incubation with rTdT nucleotide mix at 37°C for 1 hr and nuclear staining with Hoechst. All imagings were performed on a Zeiss LSM 800 confocal microscope and images were converted using ImageJ (NIH).

## Mammalian expression constructs + transfection

For mammalian cell expression, full length VPA0226 was amplified using primers 5' GATC AGATCT ATGATGAAAAAAACAATCACACTA 3' and 5' GATA GAATTC GAAACGGTACTCGGCTAAGTTGTT 3' and cloned in frame with GFP into the expression vector sfGFP-N1 (*Pédelacq et al., 2006*) (Addgene) between BglII and EcoRI sites for the expression of C terminally tagged VPA0226. Generating a catalytically inactive S153A for this construct, however, did not result in successful mammalian cell expression through transfection. Therefore, a second catalytically inactive C terminally tagged VPA0226 construct was then generated by subjecting the full-length plasmid to site-directed mutagenesis to change a single amino acid residue histidine at 393 position to alanine using the primers 5' tgagttgctgttgttggagccgtgacatcccagaacac 3' and 5' gtgttctgggatgtcacggctccaacaacagcaactca 3'. For mammalian cell transfections HeLa or HEK293T cells were plated in six-well plates containing sterile coverslips at $7 \times 10^4$ cell/ml or $2.5 \times 10^5$ cell/ml respectively and grown for 24 hr. Transient transfections were carried out for 14 hr with either 500 ng of construct DNA + 500 ng of pSFFV as filler using FugeneHD (Promega) or 1 μg of construct DNA using Polyjet transfection reagent (Fisher Scientific) according to manufacturer's instructions.

## Lipidomics

For lipidomics analysis, transfections were carried out following a previously described protocol (*Nawabi et al., 2008*). HeLa cells ($3 \times 10^6$) were plated and transfected with 2 μg of plasmids encoding VPA0226-WT sf-GFPN1 or VPA0226-H/A sf-GFPN1 using Polyjet transfection reagent according to manufacturer's instructions. 14 hr post-transfection cells were washed three times with PBS and lipids were extracted from cell pellets following Bligh and Dyer's protocol using chloroform: methanol: 0.88% NaCl in 1:2:0.8 ratio (*Bligh and Dyer, 1959*). The extracted lipids were dissolved in 15 μl of chloroform and processed for TLC analysis using plastic backed silica gel 60F254 plates (Sigma) and a mobile phase system containing petroleum ether: ethyl ether: acetic acid in 90:10:1 ratio. Lipid spots were visualized by exposure to iodine vapor. Cholesterol, cardiolipin (1',3'-bis[1,2-dioleoyl-sn-glycero-3-phospho]-glycerol), DOPS (1,2-dioleoyl-sn-glycero-3-phospho-L-serine) (Avanti Polar Lipids) and cholesteryl ester (cholesterol oleate, Sigma) were spotted as standards on the TLC plate.

## Analysis of lipids by LC-MS

All solvents were either HPLC or LC/MS grade and purchased from Sigma-Aldrich (St Louis, MO, USA). All lipid extractions were performed in $16 \times 100$ mm glass tubes with PTFE-lined caps (Fisher Scientific, Pittsburg, PA). Glass Pasteur pipettes and solvent-resistant plasticware pipette tips (Mettler-Toledo, Columbus, OH) were used to minimize leaching of polymers and plasticizers.

Samples were transferred to glass tubes for liquid-liquid extraction (LLE) by modified Bligh/Dyer (*Bligh and Dyer, 1959*); 1 mL each of dichloromethane, methanol, and water were added to the glass tube containing the sample. The mixture was vortexed and centrifuged at $2671 \times g$ for 5 min, resulting in two distinct liquid phases. The organic phase was collected to a fresh glass tube with a Pasteur pipette and dried under $N_2$. Samples were resuspended in hexane.

Lipids were analyzed by LC-MS/MS using a SCIEX QTRAP 6500$^+$ equipped with a Shimadzu LC-30AD (Kyoto, Japan) HPLC system and a $150 \times 2.1$ mm, 5 μm Supelco Ascentis silica column (Bellefonte, PA). Samples were injected at a flow rate of 0.3 ml/min at 2.5% solvent B (methyl tert-butyl ether) and 97.5% Solvent A (hexane). Solvent B was increased to 5% during 3 min and then to 60% over 6 min. Solvent B was decreased to 0% during 30 s while Solvent C (90:10 (v/v) Isopropanol-water) was set at 20% and increased to 40% during the following 11 min. Solvent C was increased to 44% during 6 min and then to 60% during 50 s. The system was held at 60% of solvent C during 1 min prior to re-equilibration at 2.5% of solvent B for 5 min at a 1.2 mL/min flow rate. Solvent D (95:5 (v/v) Acetonitrile-water with 10 mM Ammonium acetate) was infused post-column at 0.03 ml/min. Column oven temperature was set to 25°C. Data was acquired in positive and negative modes by employing polarity switching during the analysis. Data were acquired using multiple reaction

monitoring (MRM) transition for the major, abundant lipid species from all lipid classes. Electrospray ionization source parameters were, GS1 40, Cur 20, source temperature 150°C, declustering potential 60, and collision energy 25. GS1 and 2 were zero-grade air while Cur and CAD gas was nitrogen. MRM transitions for cholesteryl esters were analyzed using MultiQuant software (SCIEX).

## VPA0226 protein purification

One 6 mL culture each of CAB2Δvpa0226 transformed with pBAD 6His::VPA0226 and CAB2Δvpa0226 transformed with pBAD 6His::VPA0226 S153A was grown shaking overnight at 30°C. Overnight cultures were each used to inoculate 500 mL MLB, which was grown shaking at 30°C for 3 hr to an OD600 = 0.85. Cultures were induced in 0.1% arabinose 8–12 hr shaking at 30°C. Inductions were pelleted by centrifugation at 4°C in a JA10 Beckman and Coulter and supernatant was collected. 6xHis-tagged VPA0226 and VPA0226 S153A were purified from the supernatant using Ni2+ affinity purification.

## Phospholipase two assay

Purified VPA0226 WT and VPA0226 S/A proteins were tested for phospholipase2 (PLA2) activity using EnzCheck phospholipase A2 assay kit according to manufacturer's instructions. Briefly, 0, 0.1, 0.3 and 1 μg of protein was mixed with the proprietary substrate BODIPY PC-A2 in a 100 μl reaction buffer of 50 mM Tris-HCl, 100 mM NaCl, and 1 mM CaCl$_2$, pH 8.9 in a clear bottom, black 96-well plate and incubated at RT for 30 min, protected from light. Fluorescence was measured using a Clariostar plus plate reader with excitation/emission wavelengths at 460/515. PLA2 from bee venom provided in the kit was used as a positive control.

## Liposome assay

Large unilamellar vesicle liposomes containing 50 μMol each of cholesterol:cardiolipin (1',3'-bis[1,2-dioleoyl-sn-glycero-3-phospho]-glycerol) or 50 μMol each of cholesterol:DOPS (1,2-dioleoyl-sn-glycero-3-phospho-L-serine) (Avanti Polar Lipids) were prepared as described previously (*Sreelatha et al., 2013*). Briefly, dried lipid films containing lipid mixture were hydrated in 100 μL of 50 mM Tris, pH 7.4, 160 mM KCl by continuous vortexing for 5 min and taken through five cycles of freeze- thawing in liquid nitrogen. The resulting liposomes were extruded using an Avanti miniextruder and 100 nm polycarbonate membranes to obtain uniform liposomes. 10 μL of above liposomes were mixed with 5 μg of purified VPA0226 WT or S153A protein in 50 mM Tris, pH 7.4, 160 mM KCl plus 1.4% fat free albumin and incubated at 37°C for 1 hr (*Buckley, 1982*). Reactions were terminated by the addition of chloroform:methanol (2:1 v/v) and the lipids were extracted by Bligh and Dyers method (*Bligh and Dyer, 1959*) and dissolved in 15 μL of chloroform for TLC analysis. Lipids were separated by TLC using plastic backed silica gel H plates (Sigma) and a mobile phase system containing petroleum ether: ethyl ether: acetic acid in 90:10:1 ratio. Lipid spots were visualized by exposure to iodine vapor. Cholesterol oleate (Sigma) and above-mentioned lipids were spotted as standards on the TLC plate.

## qPCR analysis

HeLa cells were transfected with empty vector sf-GFPN1, VPA0226-WT sf-GFPN1 or VPA0226-H/A sf-GFPN1 as described above. 14 hr post transfection, RNA was isolated from the transfected cells using RNAeasy plus mini kit (Qiagen) according to the manufacturer's instructions. cDNA was generated using the qScript cDNA synthesis kit (Qunatabio). qPCR analysis for gene expression levels was carried out on a CFX384 Touch Real-Time PCR Detection System using PerfeCTa SYBR Green Supermix (Quantabio) and 500 nM primers. The primers were designed based on the established database, Primer Bank, PCR primers for Gene Expression Detection and Quantification and were checked for efficiency (*Spandidos et al., 2010*). Following sets of primers were used for the qPCR assay, all given in 5' to 3'. HMGCR: TGATTGACCTTTCCAGAGCAAG, CTAAAATTGCCATTCCACGAGC; HMGCS: CATTAGACCGCTGCTATTCTGTC, TTCAGCAACATCCGAGCTAGA; DHCR24: GCACAGGCATCGAGTCATCAT, GTGCATCGCACAAAGCTGC; GAPDH: GGAGCGAGATCCCTCCAAAAT, GGCTGTTGTCATACTTCTCATGG. Relative gene expression level for the target genes was calculated by the ΔCq method with respect to transcript levels of the housekeeping gene GAPDH. For Bacterial RNA extraction, RNEasy protect bacteria mini kit (Qiagen) was used according

to the manufacturer's instructions and the following primer pairs (5′ to 3′) were used for qPCR analysis. VPA0226: GTGGTTGCACTCGGTGACAG, TGCCCAGTTGTAGAGCGGAA; RecA: GCTAG TAGAAAAAGCGGGTG, GCAGGTGCTTCTGGTTGAG. Relative gene expression level for the target gene was calculated by the ΔCq method with respect to transcript levels of the housekeeping gene RecA.

### Plasma membrane isolation and cholesterol quantification

HeLa cells were plated at 80–90% confluency in 150 mm dishes and transfected with 5 µg of plasmid DNA encoding VPA0226-WT sf-GFPN1 or VPA0226-H/A sf-GFPN1 using Polyjet transfection reagent as described above. 14 hr post transfection, plasma membranes were isolated from the transfected cells using subcellular fractionation and differential centrifugation following previously published protocols (*Tannu and Hemby, 2006*; *Casey et al., 2017*). Briefly, cells harvested by scraping into PBS were washed and re-suspended in HNMEK lysis buffer (20 mM HEPES pH 7.4, 50 mM NaCl, 2 mM MgCl2, 2 mM EDTA, 10 mM KCl, 50 nM EGTA, protease inhibitors) for 30 min on ice. The cell suspensions were lysed by dounce homogenizer and the homogenate was centrifuged at 800 g and 4°C for 10 min to obtain the nuclei pellet (P1). The supernatant was centrifuged at 10,000 g, 4°C for 15 min to pellet the organelle fraction (P2). The supernatant was further centrifuged at 100,000 g, 4°C for 1 hr to obtain the plasma membrane (P3) and cytosol (S3) fractions. All the pellets were washed once in HNMEK buffer and samples were collected for protein quantification and western blot analysis. Free cholesterol quantification in the plasma membrane fractions was carried out using a cholesterol quantification assay kit (Sigma, CS0005) following manufacturer's instructions.

### Propidium iodide binding assay

Susceptibility of HeLa cells to mechanical stress was assessed using propidium iodide binding (*Crowley et al., 2016*). HeLa cells transfected with empty vector sf-GFPN1, VPA0226-WT sf-GFPN1 or VPA0226-H/A sf-GFPN1 for 14 hr were harvested by scraping the cells into sterile PBS. The cells were washed twice with PBS. $1 \times 10^5$ cells were plated in triplicate in a clear bottom, black 96-well plate after they are resuspended in water containing 5 µM propidium iodide (ThermoFisher Scientific) and incubated at room temperature for 10 min. Fluorescence was immediately measured using a Clariostar plus plate reader with excitation/emission wavelengths at 530/620 nm. HeLa cells treated with 1% Triton X-100 for 5 min were used as positive control and PI binding was expressed as a percentage of Triton-treated cell binding.

### Statistical analysis and bioinformatics

All data are given as mean ± standard deviation from at least three independent experiments unless stated otherwise. Each experiment was conducted in triplicate. Statistical analyses were performed by using unpaired, two-tailed Student's t test with Welch's correction and one-way ANOVA with Turkey's multiple comparison test. A p value of $< 0.05$ was considered significant.

For sequence similarity-based CLANS clustering, the RP15 representative sequence set of the Lipase_GDSL family (Pfam database ID: PF00657) was used. Sequence redundancy was removed by CD-HIT clustering at 60% sequence identity. Similarities up to BLAST E-value 0.01 were used.

CLANS: *Frickey and Lupas, 2004*.
CD-HIT: *Huang et al., 2010*.

## Acknowledgements

We thank members of the Orth lab for their discussions and editing. We thank Dr. Arun Radhakrishnan for valuable discussions and providing cholesterol/MCD reagents. This work was funded NIH grants R01 GM115188 and RO1 GM113079, Once Upon a Time…Foundation and the Welch Foundation I-1561. Dr. Jeffery G McDonald is supported by PO1 HL20948. Dr. Jen Liou is a Sowell Family Scholar in Medical Research. Dr. Kim Orth is a Burroughs Welcome Investigator, a Beckman Young Investigator, and a W W Caruth, Jr., Biomedical Scholar with an Earl A Forsythe Chair in Biomedical Science.

## Additional information

### Competing interests
Kim Orth: Reviewing Editor, eLife. The other authors declare that no competing interests exist.

### Funding

| Funder | Grant reference number | Author |
| --- | --- | --- |
| National Institutes of Health | R01 GM115188 | Kim Orth |
| National Institutes of Health | RO1 GM113079 | Jen Liou |
| Once Upon A Time Foundation | | Kim Orth |
| Welch Foundation | I-1561 | Kim Orth |
| National Institutes of Health | PO1 HL20948 | Jeffrey G McDonald |

The funders had no role in study design, data collection and interpretation, or the decision to submit the work for publication.

### Author contributions
Suneeta Chimalapati, Conceptualization, Data curation, Formal analysis, Investigation, Visualization, Methodology, Writing - original draft, Writing - review and editing; Marcela de Souza Santos, Conceptualization, Data curation, Validation, Investigation, Visualization, Methodology, Writing - review and editing; Alexander E Lafrance, Data curation, Formal analysis, Validation, Visualization; Ann Ray, Data curation, Validation; Wan-Ru Lee, Gonçalo Vale, Data curation; Giomar Rivera-Cancel, Conceptualization, Data curation; Krzysztof Pawlowski, Conceptualization, Resources; Matthew A Mitsche, Conceptualization, Methodology; Jeffrey G McDonald, Conceptualization, Data curation, Methodology; Jen Liou, Conceptualization, Resources, Data curation; Kim Orth, Conceptualization, Resources, Formal analysis, Supervision, Funding acquisition, Investigation, Visualization, Methodology, Writing - original draft, Project administration, Writing - review and editing

### Author ORCIDs
Ann Ray (iD) http://orcid.org/0000-0001-5803-2919
Giomar Rivera-Cancel (iD) http://orcid.org/0000-0002-4909-4103
Jen Liou (iD) http://orcid.org/0000-0003-1546-3115
Kim Orth (iD) https://orcid.org/0000-0002-0678-7620

### Decision letter and Author response
Decision letter https://doi.org/10.7554/eLife.58057.sa1
Author response https://doi.org/10.7554/eLife.58057.sa2

## Additional files

### Supplementary files
• Transparent reporting form

### Data availability
All data generated or analysed during this study are included in the manuscript and supporting files.

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
