## [Decision Letter]

**Acceptance summary:**

This work proposes a novel mechanism for egress of *Vibrio parahaemolyticus* from host cells involving cholesterol esterification by type II secreted VPA0226, proposed to cause a weakening of host cell membranes, including the plasma membrane, thereby promoting *Vibrio* egress. This is a potentially exciting and novel mechanism for *Vibrio* egress, since *Vibrio parahaemolyticus* relies on a T3SS to invade host cells, but this T3SS does not support escape from the host cell. The pathogen would then fully benefit from the two types of secretion systems to invade and then to escape the cells.

**Decision letter after peer review:**

[Editors’ note: the authors submitted for reconsideration following the decision after peer review. What follows is the decision letter after the first round of review.]

Thank you for submitting your work entitled "*Vibrio* deploys Type 2 secreted lipase to esterify cholesterol with host fatty acids and mediate cell egress" for consideration by *eLife*. Your article has been reviewed by a Senior Editor, a Reviewing Editor, and three reviewers. The reviewers have opted to remain anonymous.

Our decision has been reached after consultation between the reviewers. Based on these discussions and the individual reviews below, we regret to inform you that your work will not be considered further for publication in *eLife*.

Reviewers all agreed that the work presented here is conceptually very interesting, particularly as it provides a potentially exciting and novel mechanism for pathogen egress, an important aspect of bacterial pathogenesis. However, consensus of all reviewers was that significant additional work (estimated to take many months) is required to strengthen the proposed overall conclusion of this study. Therefore, we decided to reject the manuscript for now. For guidance, the reviewers' comments are available below. We hope that you find them constructive and that they help you amend the work.

Reviewer #1:

This work proposes a novel mechanism for egress of *Vibrio parahaemolyticus* from host cells involving cholesterol esterification by type II secreted VPA0226. This is proposed to cause a weakening of host cell membranes, including of the plasma membrane, thereby promoting *Vibrio* egress. This is a potentially exciting and novel mechanism for pathogen egress, a mostly underexplored aspect of bacterial pathogenesis. However, the proposed model is essentially based in experiments performed in mammalian cells ectopically expressing VPA0226 or in biochemical experiments with purified VPA0226. The evidence to establish that the proposed egress mechanism occurs in infected cells must be strengthened.

Essential revisions:

1) It is essential to show that free cholesterol levels are indeed decreased in infected cells by biochemical analysis of host lipids and/or (at the very least) by using analysis of the expression of HMGCR, HMGCS and DHCR24 as readout.

2) Another aspect that should be better addressed is the subcellular localization of VPA0226 during host cell infection, which could aid at a better understanding of its biological role. For example, it is unclear how the localization of ectopically expressed VPA0226 at the mitochondria relates to its proposed activity and function. With the available anti-VPA0226 antibody, or by using *V. parahaemolyticus* transformed with epitope tagged VPA0226, it should be possible to monitor the presence and/or localization of the protein within infected cells by controlled protein fractionation and/or by immunofluorescence microscopy.

Reviewer #2:

Summary:

This manuscript by de Souza Santos et al. describes the characterization of VPA0226, a *Vibrio* protein required for egress of intracellular bacteria. The authors provide experiments in support of a model in which VPA0226 is secreted by the type 2 secretion system and is required for esterification of cholesterol, weakening the plasma membrane and allowing for bacterial egress. Although the absence of VPA0226 clearly leads to accumulation of bacteria inside cells at the time of observation, the presented experiments are not sufficient to unambiguously support the model proposed by the authors.

Essential revisions:

1) The exact role of the T2SS

The notion that VPA0226 is a T2SS substrate relies on Western blot experiment in which the authors analyze the presence of VPA0226 in the bacterial pellet (expression) and in the supernatant (secretion). In agreement with the notion that VPA0226 is a T2SS substrate, it is not present in the supernatant of the T2SS mutant. However, it is not present in the pellet either. It seems cautious to conclude that VPA0226 is not expressed at all in the T2SS mutant. How could the authors then conclude that VPA0226 is a T2SS substrate? Finally, if VPA0226 requires the T2SS for function, then one would expect the T2SS mutant to display an egress defect. Is it the case?

2) Cholesterol esterification and weakening of the plasma membrane

Although the authors show modification of total cholesterol, they do not demonstrate that the modified version is in the plasma membrane, as implied by their model. Also, the papers cited in reference of cholesterol esterification leading to weakening of the plasma membrane are misused. In these studies, it was not esterification but depletion of total cholesterol that was tested, and the treatment did not lead to membrane weakening, but remodeling of tight junctions. As a consequence, there is no evidence that cholesterol esterification weakens the plasma membrane and the authors need to demonstrate what remains an assumption.

3) The exact role of VPA0266 in the observed egress defect

It is suggested that VPA02266-mediated cholesterol esterification leads to plasma membrane leakiness, thus facilitating egress. If this is true, then increasing cholesterol esterification (by an independent method, using manipulation of cholesterol metabolism for instance) should rescue the bacterial egress defect of the VPA0266 mutant.

4) Biological significance of the observed defects

The authors should consider studying the role of VPA0266 in egress using various epithelial cell types. Importantly, using Caco-2 cells instead of HeLa cells would address the potential role of cell polarity in VPA0226 function (and *Vibrio* infection). Perhaps more importantly, the authors should demonstrate the role of VPA0266 in existing models of *Vibrio* pathogenesis.

Reviewer #3:

The manuscript by de Souza Santos et al., describes the characterization of the secreted lipase VPA0226 and its requirement for cellular egress by *Vibrio parahaemolyticus*. The major findings include:

- the enzymatic activity of VPA0226 is required for bacterial escape from host cells;

- the somewhat surprising discovery that the lipase is not secreted by the type 3 secretion system 2. Instead, it enters the extracellular space through the type 2 secretion system;

- VPA0226 associates with mitochondria and causes mitochondrial fragmentation;

- while apoptosis is induced, it is not required for cell egress by *V. para*;

- VPA0226 esterifies cholesterol

The results are intriguing, the cell biology is convincing and the study represents a significant advance to the field, although the mechanism by which VPA0226 induces cell egress is not fully established. While the plasma membrane is destabilized as measured by Sytox permeability and sensitivity to osmotic stress following infection by *V. para* expressing active lipase and transfection of active lipase, respectively, it is not clear if this is due to esterification of cholesterol in the plasma membrane. Are we to assume that the effect is on plasma membrane cholesterol since the majority of cellular cholesterol is found in the plasma membrane? Why then is there such a dramatic effect on mitochondria that only contains a fraction of cellular cholesterol? The authors could isolate plasma membrane before performing lipidomic analysis rather than analyzing total cellular lipids to verify that plasma membrane cholesterol is modified.

A strain of *V. para* that is deleted for hemolysins and the T3SS1 was used in this study. What is the role of VPA0226 in wild type *V. para*? Is it required for cellular escape or is this phenotype masked in the presence of other virulence factors?

VPA0226 is capable of esterifying cholesterol using cardiolipin in vitro; however, another lipid source (s) of fatty acids is used in the host cell. Please address this difference. This lipase seems to have broad specificity.

Regarding the results in Figure 3A: "a clear difference" was not observed "in the migration pattern of total lipids from VPA0226 WT transfected compared to those transfected with empty vector and the inactive VPA0226 HA mutant" (subsection “VPA0226 esterifies cholesterol using host polyunsaturated fatty acids”). Please modify text.

The inactive VPA0226 that has the catalytic residue serine-153 substituted with alanine (VPA0226 S/A) was used in the infection experiments, while the transfection experiments used VPA0226 with a His393Ala substitution (VPA0226 H/A). VPA0226 S/A was purified and shown to lack acyl-transferase activity; however, that VPA0226 H/A is inactive was not verified. The activity of this variant should be determined. That VPA0226 S/A could not be expressed in the mammalian cells and an alternative mutant VPA0226 was made for the transfection experiments should be mentioned in the Results section.

The statistically significant reduction in entry into HeLa cells by *V. para* lacking VPA0226 was not addressed (Figure 1A). Is this restored with complementation? Does VPA0226 affect the plasma membrane from the outside?

[Editors’ note: further revisions were suggested prior to acceptance, as described below.]

Thank you for submitting your article "*Vibrio* deploys Type 2 secreted lipase to esterify cholesterol with host fatty acids and mediate cell egress" for consideration by *eLife*. Your article has been reviewed by Wendy Garrett as the Senior Editor, a Reviewing Editor, and three reviewers. The reviewers have opted to remain anonymous.

The reviewers have discussed the reviews with one another extensively and the Reviewing Editor has drafted this decision to help you prepare a revised submission.

The editors have judged that your manuscript is of significant interest, but as described below additional experiments are required before it is published. Because many researchers have temporarily lost access to their labs given the COVID19-pandemic, we will give authors as much time as they need to submit revised manuscripts. We are also offering, if you choose, to post the manuscript to bioRxiv (if it is not already there) along with this decision letter and a formal designation that the manuscript is "in revision at *eLife*". Please let us know if you would like to pursue this option. (If your work is more suitable for medRxiv, you will need to post the preprint yourself, as the mechanisms for us to do so are still in development.)

This work proposes a novel mechanism for egress of *Vibrio parahaemolyticus* from host cells involving cholesterol esterification by type II secreted VPA0226, proposed to cause a weakening of host cell membranes, including the plasma membrane, thereby promoting *Vibrio* egress. This is a potentially exciting and novel mechanism for *Vibrio* egress, since *Vibrio parahaemolyticus* relies on a T3SS to invade host cells, but this T3SS does not support escape from the host cell. The pathogen would then fully benefit from the two types of secretion systems to invade and then to escape the cells. The reviewers have all reviewed a previous version of this article, and all agreed that the work had been significantly improved. There remains a contentious point however about the fact that VPA0226 is a T2SS substrate and reviewers propose a list of experiments to unambiguously demonstrate the claim:

1) Because the VPA0226 protein cannot be detected in the pellet of the T2SS mutant, it is hard to definitely conclude that the T2SS is required for secretion, as opposed to any indirect effect including expression or protein stability. It is quite possible that there is a global effect on gene expression in the T2SS mutant since this mutant is unable to invade host cells.

Please test by qPCR if the loss of intracellular VPA0226 detection is due to loss of expression. Then please express VPA0226 from a plasmid in the WT and T2SS mutant that allows for inducible expression and try to detect periplasmic accumulation of VPA0226. In the end, pulse-chase experiments to determine whether VPA0226 gets degraded in the periplasm of the T2SS mutant may be necessary.

2) Please address the question of timing of egress that you commented on were halted in the mutant as opposed to delayed with the bacteria ultimately dying in intact cells. To ensure that bacterial death is not due to gentamicin in the medium ultimately reaching intracellular bacteria, please repeat the experiment, killing the extracellular bacteria with gentamicin for an hour after invasion then replacing with fresh medium and very low levels of gentamicin. It would add to the paper if cytosolic bacteria could egress in the end, through host cell exhaustion, but *Vibrio* has evolved a very specific way of doing it.

---

## [Author Response]

[Editors’ note: the authors resubmitted a revised version of the paper for consideration. What follows is the authors’ response to the first round of review.]

Reviewer #1:[…]Essential revisions:1) It is essential to show that free cholesterol levels are indeed decreased in infected cells by biochemical analysis of host lipids and/or (at the very least) by using analysis of the expression of HMGCR, HMGCS and DHCR24 as readout.

We agree that infected cells should be tested for changes in levels of free cholesterol. To test for VPA0226-mediated changes in levels of cholesterol in infected cells, we treated CAB2 infected host cells with 100µM cholesterol resulting in the addition of free cholesterol to plasma membrane and discovered that CAB2 was no longer able to efficiently egress from cholesterol treated cells. Similarly, we treated CAB2Δvpa0226 infected cells with hydroxypropyl βcyclodextrin (HPCD), a compound that depletes cholesterol from membranes, and observed that CAB2Δvpa0226 could now escape from host cells. These studies support our model that changes in free cholesterol are important for egress of *V. para* from host cells.

2) Another aspect that should be better addressed is the subcellular localization of VPA0226 during host cell infection, which could aid at a better understanding of its biological role. For example, it is unclear how the localization of ectopically expressed VPA0226 at the mitochondria relates to its proposed activity and function. With the available anti-VPA0226 antibody, or by using V. parahaemolyticus transformed with epitope tagged VPA0226, it should be possible to monitor the presence and/or localization of the protein within infected cells by controlled protein fractionation and/or by immunofluorescence microscopy.

We tried to visualize VPA0226 in infected cells both with the antibody for VPA0226 and a VPA0226-3XFLAG tagged protein, but were unable to visualize VPA0226 by immunostaining.

Reviewer #2:[…]Essential revisions:1) The exact role of the T2SSThe notion that VPA0226 is a T2SS substrate relies on Western blot experiment in which the authors analyze the presence of VPA0226 in the bacterial pellet (expression) and in the supernatant (secretion). In agreement with the notion that VPA0226 is a T2SS substrate, it is not present in the supernatant of the T2SS mutant. However, it is not present in the pellet either. It seems cautious to conclude that VPA0226 is not expressed at all in the T2SS mutant. How could the authors then conclude that VPA0226 is a T2SS substrate? Finally, if VPA0226 requires the T2SS for function, then one would expect the T2SS mutant to display an egress defect. Is it the case?

The westerns for these experiments have been repeated with proper controls. The absence of a secreted protein in the pellet fraction may indicate that the enzyme may be toxic to the bacteria. In support of this hypothesis, we tried to express VPA0226 without its secretion signal and observed no protein in the bacterial pellet or the supernatant. *V. para* containing a T2SS mutant is unable to invade host cells, which could be explained by the lack of other unexplored T2SS factors that may be involved in the holistic process of invasion into a host cell (data not shown).

2) Cholesterol esterification and weakening of the plasma membraneAlthough the authors show modification of total cholesterol, they do not demonstrate that the modified version is in the plasma membrane, as implied by their model. Also, the papers cited in reference of cholesterol esterification leading to weakening of the plasma membrane are misused. In these studies, it was not esterification but depletion of total cholesterol that was tested, and the treatment did not lead to membrane weakening, but remodeling of tight junctions. As a consequence, there is no evidence that cholesterol esterification weakens the plasma membrane and the authors need to demonstrate what remains an assumption.

We agree that infected cells should be tested for changes in levels of free cholesterol on the plasma membrane. To test for VPA0226-mediated changes in levels of cholesterol in infected cells, we treated CAB2 infected host cells with 100µM cholesterol resulting in the addition of free cholesterol to plasma membrane and discovered that CAB2 was no longer able to efficiently egress from Cholesterol treated cells. Similarly, we treated CAB2Δvpa0226 infected cells with hydroxypropyl β-cyclodextrin (HPCD), a compound that depletes cholesterol from membranes, and observed that CAB2Δvpa0226 could now escape from host cells. These studies support our model that changes in free cholesterol are important for egress of *V. para* from host cells.

We have also added new data wherein we have isolated plasma membrane fractions from HeLa cells transfected with VPA0226 or catalytically inactive VPA0226 H/A and observed that there was reduced free cholesterol content in cells expressing wild-type VPA0226 compared to those of the cells expressing either empty vector or catalytically inactive VPA0226 (Figure 5B). We have modified text to reflect the references cited.

3) The exact role of VPA0266 in the observed egress defectIt is suggested that VPA02266-mediated cholesterol esterification leads to plasma membrane leakiness, thus facilitating egress. If this is true, then increasing cholesterol esterification (by an independent method, using manipulation of cholesterol metabolism for instance) should rescue the bacterial egress defect of the VPA0266 mutant.

As suggested and described above, we treated CAB2Δvpa02226 infected cells with hydroxypropyl β-cyclodextrin (HPCD), a compound that depletes cholesterol, and observed that CAB2Δvpa0226 could now escape from host cells.

4) Biological significance of the observed defectsThe authors should consider studying the role of VPA0266 in egress using various epithelial cell types. Importantly, using Caco-2 cells instead of HeLa cells would address the potential role of cell polarity in VPA0226 function (and Vibrio infection). Perhaps more importantly, the authors should demonstrate the role of VPA0266 in existing models of Vibrio pathogenesis.

As suggested by this reviewer, we infected Caco-2 cells, and observed egress of CAB2 but not CAB2Δvpa0226 at 7 hours post gentamicin treatment (Figure 1—figure supplement 1E).

Reviewer #3:The manuscript by de Souza Santos et al., describes the characterization of the secreted lipase VPA0226 and its requirement for cellular egress by Vibrio parahaemolyticus. The major findings include:• the enzymatic activity of VPA0226 is required for bacterial escape from host cells;• the somewhat surprising discovery that the lipase is not secreted by the type 3 secretion system 2. Instead, it enters the extracellular space through the type 2 secretion system;• VPA0226 associates with mitochondria and causes mitochondrial fragmentation;• while apoptosis is induced, it is not required for cell egress by V. para;• VPA0226 esterifies cholesterolThe results are intriguing, the cell biology is convincing and the study represents a significant advance to the field, although the mechanism by which VPA0226 induces cell egress is not fully established. While the plasma membrane is destabilized as measured by Sytox permeability and sensitivity to osmotic stress following infection by V. para expressing active lipase and transfection of active lipase, respectively, it is not clear if this is due to esterification of cholesterol in the plasma membrane. Are we to assume that the effect is on plasma membrane cholesterol since the majority of cellular cholesterol is found in the plasma membrane? Why then is there such a dramatic effect on mitochondria that only contains a fraction of cellular cholesterol? The authors could isolate plasma membrane before performing lipidomic analysis rather than analyzing total cellular lipids to verify that plasma membrane cholesterol is modified.

We observe fragmentation of the mitochondria that is consistent with esterification of cholesterol and the activation of SREBP signaling, indicative of decreased cholesterol in the cell. We have isolated plasma membrane fractions from HeLa cells transfected with VPA0226 or catalytically inactive VPA0226 H/A and observed that there was reduced free cholesterol content in cells expressing wild-type VPA0226 compared to those of the cells expressing either empty vector or catalytically inactive VPA0226 (Figure 5B).

A strain of V. para that is deleted for hemolysins and the T3SS1 was used in this study. What is the role of VPA0226 in wild type V. para? Is it required for cellular escape or is this phenotype masked in the presence of other virulence factors?

The invasion phenotype in tissue culture cells is masked by the T3SS1 and the hemolysins. Trying to visualize this phenotype in animals is out of the scope of this study.

VPA0226 is capable of esterifying cholesterol using cardiolipin in vitro; however, another lipid source(s) of fatty acids is used in the host cell. Please address this difference. This lipase seems to have broad specificity.

This lipase does have broad specificity. In vitro, as this is a bacterial lipase, cardiolipin was the logical substrate to use for transferase experiments. However, we observe that when given a choice of all phospholipids in a host cell, the lipids transferred were 20:3, 20:4, 20:5 and 22:6 to induce the production of CE(20:3), CE(20:4), CE(20:5) and CE(22:6).

Regarding the results in Figure 3A: "a clear difference" was not observed "in the migration pattern of total lipids from VPA0226 WT transfected compared to those transfected with empty vector and the inactive VPA0226 HA mutant" (subsection “VPA0226 esterifies cholesterol using host polyunsaturated fatty acids”). Please modify text.

The text has been modified and we do observe esterified cholesterol based on our lipidomic experiments (Figure 3A). As the TLC lipidomic analysis is not quantitative and we cannot identify the particular cellular lipid used in the by VPA0226 and, as suggested by other reviewers, we have decided to delete the cellular TLC experiments previously shown in Figure 3A and Figure 3—figure supplement 2C.

The inactive VPA0226 that has the catalytic residue serine-153 substituted with alanine (VPA0226 S/A) was used in the infection experiments, while the transfection experiments used VPA0226 with a His393Ala substitution (VPA0226 H/A). VPA0226 S/A was purified and shown to lack acyl-transferase activity; however, that VPA0226 H/A is inactive was not verified. The activity of this variant should be determined. That VPA0226 S/A could not be expressed in the mammalian cells and an alternative mutant VPA0226 was made for the transfection experiments should be mentioned in the Results section.

We have mentioned that we used two different mutants and why in the Results section.

Since our S153A mutant was unable to be stably express in mammalian cells we have used a Histine 393 Alanine mutant that mutates another residue in the conserved catalytic triad that previously was documented (Akoh and Shaw et al., 2004). The H393A mutant when tested in gentamicin protection assay behaved similar to the S153A mutant. (Figure 4—figure supplement 2).

The statistically significant reduction in entry into HeLa cells by V. para lacking VPA0226 was not addressed (Figure 1A). Is this restored with complementation?

We have since repeated these assays and found that the entry of the *V. para* lacking VPA0226 was not reduced in a statically significant way. We apologize for this mis-judgment and have updated the figure to reflect this point.

Does VPA0226 affect the plasma membrane from the outside?

VPA0226 seems to affect only invaded cells, as we observe mitochondrial fragmentation only in invaded cells. Data for this observation was added.

[Editors’ note: what follows is the authors’ response to the second round of review.]

[…] There remains a contentious point however about the fact that VPA0226 is a T2SS substrate and reviewers propose a list of experiments to unambiguously demonstrate the claim:1) Because the VPA0226 protein cannot be detected in the pellet of the T2SS mutant, it is hard to definitely conclude that the T2SS is required for secretion, as opposed to any indirect effect including expression or protein stability. It is quite possible that there is a global effect on gene expression in the T2SS mutant since this mutant is unable to invade host cells.Please test by qPCR if the loss of intracellular VPA0226 detection is due to loss of expression. Then please express VPA0226 from a plasmid in the WT and T2SS mutant that allows for inducible expression and try to detect periplasmic accumulation of VPA0226. In the end, pulse-chase experiments to determine whether VPA0226 gets degraded in the periplasm of the T2SS mutant may be necessary.

As suggested by the reviewer, we performed the following experiments (subsection “VPA0226 is secreted by the type 2 secretion system”):

“As we did not observe VPA0226 in the pellet fraction of CAB2∆epsD, we performed qPCR analysis to confirm the expression of VPA0226 in CAB2∆epsD strain and observed a significant reduction in the expression level of VPA0226 in CAB2∆epsD compared to CAB2 (Figure 2C). However, when we overexpressed VPA0226 through a plasmid in CAB2∆epsD, we observed a slightly higher migrating band corresponding to the uncleaved version of VPA0226 in the pellet fraction along with a few other lower migrating bands indicating a possible degradation of VPA0226(Figure 2D). Taken together, these results demonstrate that VPA0226 is secreted by the type 2 secretion system (Figure 2E).”

2) Please address the question of timing of egress that you commented on were halted in the mutant as opposed to delayed with the bacteria ultimately dying in intact cells. To ensure that bacterial death is not due to gentamicin in the medium ultimately reaching intracellular bacteria, please repeat the experiment, killing the extracellular bacteria with gentamicin for an hour after invasion then replacing with fresh medium and very low levels of gentamicin. It would add to the paper if cytosolic bacteria could egress in the end, through host cell exhaustion, but Vibrio has evolved a very specific way of doing it.

As suggested, we performed a gentamycin protection assay with lower levels of gentamycin and confirmed that bacterial death is not due to experimental design.

Subsection “The catalytic activity of VPA0226 is required for *V. parahaemolyticus* egress from host cell”: “…we performed an extended time course gentamicin protection assay in which Hela cells were infected with *V. parahaemolyticus* strains for 2 hours, treated with gentamicin at 100µg/ml for 1 hour and then changed to 10µg/ml for the remainder of the assay (3, 6, 8,12 and 24 hours). As observed previously, Hela cells invaded with CAB2∆vpa0226 or CAB2∆vpa0226+S/A strains still contained viable bacteria even at 12h PGT in contrast to CAB2 or CAB2∆vpa0226+WT invaded Hela cells that had no viable bacteria from 8h PGT onwards (Figure 1—figure supplement 2A and B). However, at 24 hours PGT, no viable bacteria were present in HeLA cells under any conditions and majority of the Hela cells appeared rounded possibly due to exhaustion (Figure 1—figure supplement 2B).”